# Amphibious Aircraft Developments: Computational Studies of Hydrofoil Design for Improvements in Water-Takeoffs †

**Arjit Seth ‡** and **Rhea P. Liem *,‡**

Department of Mechanical and Aerospace Engineering, Hong Kong University of Science and Technology, Hong Kong 999077, China; aseth@connect.ust.hk

* Correspondence: rpliem@ust.hk

† This paper is an extended version of our paper published in Proceedings of the AIAA Aviation 2019 Forum, Dallas, TX, USA, 17–21 June 2019; p. 3552.

‡ These authors contributed equally to this work.

**Abstract:** Amphibious aircraft designers face challenges to improve takeoffs and landings on both water and land, with water-takeoffs being relatively more complex for analyses. Reducing the water-takeoff distance via the use of hydrofoils was a subject of interest in the 1970s, but the computational power to assess their designs was limited. A preliminary computational design framework is developed to assess the performance and effectiveness of hydrofoils for amphibious aircraft applications, focusing on the water-takeoff performance. The design framework includes configuration selections and sizing methods for hydrofoils to fit within constraints from a flying-boat amphibious aircraft conceptual design for general aviation. The position, span, and incidence angle of the hydrofoil are optimized for minimum water-takeoff distance with consideration for the longitudinal stability of the aircraft. The analyses and optimizations are performed using water-takeoff simulations, which incorporate lift and drag forces with cavitation effects on the hydrofoil. Surrogate models are derived based on 2D computational fluid dynamics simulation results to approximate the force coefficients within the design space. The design procedure is evaluated in a case study involving a 10-seater amphibious aircraft, with results indicating that the addition of the hydrofoil achieves the purpose of reducing water-takeoff distance by reducing the hull resistance.

**Keywords:** amphibious aircraft; hydrofoils; takeoff performance; computational fluid dynamics; optimization

## 1. Introduction

Amphibious aircraft have the potential to play an important role in passenger transport as part of general aviation, particularly in short-range flights [1]. Aircraft designed for takeoff and landings solely on runways are limited by the number of airports present in regions. The use of water bodies and ports as additional takeoff and landing points pose larger versatility and scope for missions with the use of amphibious aircraft. Mission profiles, weight estimation, fuel efficiency, range, payload, and stability are key considerations in the preliminary design and development of any aircraft. However, the takeoff and landing flight segments for amphibious aircraft present greater complications than ones in general aviation aircraft design because of the increased complexities involving water analyses. These aircraft suffer from several restrictions in analyses, due to the complexities of characteristic aerodynamic and hydrodynamic parameters, that make it difficult to non-dimensionalize and test models with respect to a length scale; these complexities are described in Section 2.1.

Amphibious aircraft are mostly used as utility vehicles for search and rescue operations, payload delivery in remote and undeveloped areas, and firefighting efforts, among others. Sikorsky Aircraft developed numerous amphibious aircraft from the mid-1920s till 1940 for varied customers (Popular Aviation, October 1931: https://books.google.com/books?id=

qZhdxqqWu4gC&pg=PA89). The Piaggio P. 136 was an amphibious aircraft used by the Italian Air Force in the 1950s (http://www.aeroflight.co.uk/waf/italy/af/ital-af2-all-time.htm). More recently, the ShinMaywa US-2 has been in production, with a history of amphibious aircraft development since the 1950s (https://www.shinmaywa.co.jp/aircraft/english/us2/us2_history.html). The Dornier Seastar is an amphibious aircraft from the 1970s with a new generation currently under development (https://www.flightglobal.com/business-aviation/dorniers-new-generation-seastar-makes-maiden-sortie/137601.article). The AVIC AG-600 "Kunlong", currently under development in China, is claimed to be the largest amphibious aircraft developed for aerial firefighting, maritime patrol, and search and rescue operations (http://www.xinhuanet.com/english/2018-05/13/c_137176292.htm). The research on amphibious aircraft development is lacking partly due to its focus on specific missions for utility purposes [2], as well as the increased complexities of hull and float analyses. The early developments of amphibious aircraft relied mainly on experimental water tank tests and empirical methods. As such, a thorough design exploration and optimization was not possible due to computational limitations [3,4]. A common strategy is to take conventional aircraft and convert them into amphibious configurations by the addition of a hull or float, which results in suboptimal designs. The aim for truly optimal amphibious aircraft design would require a systematic design procedure that considers preliminary sizing, stability and control, and shape optimization. A conceptual design and sizing framework for amphibious aircraft was developed by Cary [5]. While numerical optimization has been commonly used in aircraft design since the late 1970s [6,7], its application in amphibious aircraft is still limited. Qiu and Song recently performed a response-surface-based multiobjective optimization to find the optimum hull step configuration, achieving an 18% improvement in takeoff distance without sacrificing cruise performance substantially [8]. Puorger et al. performed an aerostructural optimization for a fire-extinguisher amphibious aircraft [9], focusing mainly on the ground effect during low-altitude cruise, without any water-takeoff considerations. A water-takeoff performance calculation method using low-cost empirical models has also been recently developed by Wang et al. [10]; this method was derived based on digital virtual flight, and included a pilot model to study the impact of a lack of visual references during water-takeoff. The model was validated against some experimental data with less than 10% error.

A particular technology for the potential improvement of amphibious aircraft water-takeoff performance is the hydrofoil. In this paper, a hydrofoil is defined as a lifting surface that travels through water. The implementation of the technology allows a ship's hull to reach a planing stage more quickly, and effectively unport the hull from water. This reduces motor effort by reducing hull resistance, which allows it to travel at higher speeds. They have been researched and implemented in water-based craft since the late 1800s, with extensive research projects performed between the 1930s and the 1960s to improve performance of marine vehicles [11]. Experiments showing the effectiveness of hydrofoils dated as early as 1861 by Thomas Moy [12], in 1907 and 1914 by the Wright brothers [3], and in 1937–1940 by Sottorf [13], among many others. Most modern studies of hydrofoils relate to marine and naval technologies, including unmanned surface vehicles (USV) [14–16], aquatic unmanned aerial vehicle (AquaUAV) [17], turbomachinery [18], naval propellers [19], the propulsion of marine vehicles [20,21], or axial-flow pumps [22]. Hydrofoil boats, such as the "Raketa" developed in Russia in 1957, were manufactured and used for commercial transporation [23]. Studies of hydro-aerodynamic characteristics of hydrofoil systems in vessels were performed by Plisov and Rozhdestvenski [24]. A hydrofoil, as opposed to an airfoil, can suffer from cavitation and ventilation [25], further described in Section 2.2. Shen and Eppler researched various hydrofoil profiles using inverse design methods for U.S. hydrofoil craft [26]. Studies of two-phase boundary layer control in cavitating and supercavitating conditions were performed by La Roche and Trevisani in relation to the Supramar company [27]. Garg et al. performed multipoint hydrodynamic shape and hydrostructural optimizations of 3D hydrofoils in partially cavitating conditions to improve fuel efficiency of ships [28]. This analysis, however, did not consider multiphase flows

and enforced a cavitation constraint based on the pressure coefficient to obtain a shape that did not undergo cavitation, which might not be suitable for high-speed conditions. Vernengo et al. performed shape optimizations of 2D supercavitating hydrofoils with multiphase flows for the development of high-speed planing craft [29].

Amphibious aircraft with hulls can also benefit from hydrofoils similarly by reaching their takeoff speeds more quickly, reducing the distance and time in the water-takeoff stage. The research on hydrofoils for amphibious aircraft design is sparse; the number of designed hydrofoil profiles is far lesser compared to that of airfoils, and the number of investigations into hydrofoil shape optimization with the relevant constraints is much lesser, as well. The Thurston Aircraft Corporation published a report which extensively considered hydrofoils for seaplane design [3] based on experimental results, but not for amphibious aircraft with the relevant constraints. David Thurston also investigated general amphibious aircraft design with minor focus on the implementation of a hydrofoil in a working concept called the HRV-1 flown in 1974, but the rigor of analysis for the takeoff condition was limited to a basic sizing and did not consider multiple configurations [2]. Cary applied Thurston's procedure into the amphibious aircraft design framework with hydrofoil considerations, but the scope of foil profile selection, viability of design, and stability were not investigated [5]. The hydrodynamic lift for flying boats and seaplanes via the use of hydrofoils for a particular retractable configuration was invented by Zimmer and patented under Dornier Luftfahrt [30]. The LISA Akoya (LISA Airplanes: http://lisa-airplanes.com/en/light-amphibious-aircraft-akoya/), which is a two-seater aircraft designed for leisure flight, is an amphibious aircraft known to implement this technology in its design.

A rigorous computational framework that can systematically evaluate and assess the impact of hydrofoil performance in amphibious aircraft is still lacking. The numerical studies on amphibious aircraft performance mentioned above [8–10] did not consider hydrofoils and thus would not be directly applicable to amphibious aircraft with hydrofoils. Such a framework will enable tradeoff studies and optimization, which can help bring revolutionary technological improvement to amphibious aircraft design, e.g., by optimizing the shape of the hull and hydrofoils. The current investigation aims to formulate a computational design framework for preliminary hydrofoil design in the context of amphibious aircraft application. This will include procedures for hydrofoil sizing and placement, with longitudinal stability considerations. One goal of the implementation is to provide a "riding" surface for amphibious aircraft while in the planing stage to reduce hull drag by minimizing its water contact. In particular, we will investigate and compare the performance of amphibious aircraft hull with and without hydrofoils in different speed regimes during the water-takeoff procedure. The results will give us physical insights into the design problem.

The paper is structured as follows. Section 2 discusses the framework of fluid dynamics important in amphibious aircraft design, including hydrofoils. Section 3 outlines the research methodology, including the sizing procedure of the hydrofoil, the explanation of the water-takeoff model for amphibious aircraft, the considerations of the effects of the hydrofoil on the stability of the aircraft, and the design framework for the analysis and optimization of the hydrofoil. Section 4 discusses the application of the hydrofoil design framework to a 10-seater amphibious aircraft design with its results. The summary of key findings and conclusion of this work are then presented in Section 5.

## 2. Theory and Hydrofoil Design Challenges

### 2.1. Hydrodynamic Analyses

Three key non-dimensional parameters in water analysis are the Reynolds ($Re$), Froude ($Fr$), and Weber ($We$) numbers:

$$Re \equiv \frac{\rho u L}{\mu}, \quad Fr \equiv \frac{u}{\sqrt{gL}}, \quad We \equiv \frac{\rho u^2 L}{\kappa}, \tag{1}$$

where $\rho$ is the density of the fluid, $u$ is the freestream speed, $L$ is the reference length of the body of analysis, $\mu$ is the dynamic viscosity, $g$ is the gravitational acceleration, and $\kappa$ is the surface tension between the two fluids under consideration. Fluid parameters corresponding to air will be denoted by a subscripted $A$, e.g., $\mu_A$, and similarly $W$ for water. The Weber number does not scale accordingly with Reynolds and Froude numbers because it goes as the square of the velocity, so the benefits of non-dimensional analysis are lost when attempting to size a water-based component of a transport vehicle based on a scale model with experimental results. The case worsens with the Froude number going as the reciprocal of the square root of the length against the Reynolds numbers' proportional relationship to length. The mitigating strategy is to use the same Froude number in the model tests and to adjust for different Reynolds numbers when scaling. Some errors exist in water spray, wave pattern and foaming predictions due to the difference in Weber numbers, but these are negligible in resistance prediction of scaled-up hulls [31].

A preliminary calculation under assumptions of constant viscosity and density shows that the Reynolds number in fresh water is approximately 16 times greater than the Reynolds number in air for the same speed and reference length. Lift and drag forces in fresh water are approximately 815 times ($\rho_W/\rho_A \approx 840$ at International Standard Atmosphere sea level conditions with salinity considerations) greater for foils of the same profile and dimensions traveling at a given speed with completely attached flow in both cases. Turbulence, cavitation, and ventilation effects correspondingly result in non-trivially determined differences between the two mediums.

Hydrodynamic forces important in hull design, such as resistance and buoyant force, are non-dimensionalized by division with $\rho_W g B^3$, where $B$ is the hull width, as opposed to the product of the dynamic pressure and wing area in aerodynamics; this is usually because the hull lengths are fixed and the widths are varied for design analyses. Archimedes' principle also justifies the inclusion of gravitational acceleration as a term to account for buoyancy forces.

The following hydrodynamic coefficients, called the load, resistance, and velocity coefficients ($C_\Delta$, $C_R$, and $C_V$, respectively), play an important role in the water-takeoff analysis of an amphibious aircraft with a hull as non-dimensional measures of buoyancy $\Delta$, resistance $R$, and speed $u$:

$$C_\Delta \equiv \frac{\Delta}{\rho_W g B^3}, \quad C_R \equiv \frac{R}{\rho_W g B^3}, \quad C_V \equiv \frac{u}{\sqrt{gB}}. \tag{2}$$

The trim angle $\alpha_{\text{trim}}$ of a hull is defined as the angle at which a boat must be longitudinally inclined for a given speed with respect to its orientation at rest such that it maintains optimal performance via minimization of resistance generated. The resistance and trim angle of a hull are experimentally determined with variations against speed via tank tests of models, which are then scaled using methods, such as one from the International Towing Tank Conference in 1978 (ITTC-78) [31].

The non-dimensional lift $C_L$, drag $C_D$, and moment $C_M$ coefficients that characterize airfoil performance are also used for hydrofoils with the freestream density and dynamic viscosity of water as reference. However, phase changes can occur over lifting bodies traveling in water at speeds within the takeoff speed regime of aircraft, leading to additional complications not seen in airfoils. Here, we use the generic non-dimensionalization for the lift $L$, drag $D$, and moment $M$:

$$C_L \equiv \frac{L}{\frac{1}{2}\rho u^2 S}, \quad C_D \equiv \frac{D}{\frac{1}{2}\rho u^2 S}, \quad C_M \equiv \frac{M}{\frac{1}{2}\rho u^2 S c}, \tag{3}$$

where $\rho$, $u$, $S$, $c$ refer to density, velocity, area, and chord of the lifting component, respectively. The subscripts $hf$, $h$, and $w$ will denote the parameters corresponding to hydrofoil, horizontal tail, and wing, respectively.

### 2.2. Cavitation and Ventilation

Cavitation and ventilation can notably reduce hydrodynamic efficiency of hydrofoils. Cavitation occurs when the local pressure approaches the saturated vapor pressure [28], which can cause flow-induced noise and vibration, decreased lift and thrust, and increased drag [32]. Ventilation refers to the phenomenon where air bubbles are found in the submerged part of the body [15], and this can result in a loss of lift [3,16]. The measure of ventilation effects is considered to be the depth of submergence of the lifting surface from the free surface. In the case of a ventilation analysis for a hydrofoil, the depth of submergence in water from sea level is the appropriate measure. Early work to optimize the hydrodynamics of hydrofoils did not consider cavitation [33], though most subsequent works considered it [18,20,34–36]. These works mainly used potential flow as their bases. In recent decades, some studies have used high-fidelity models, in particular the Unsteady Reynolds-averaged Navier–Stokes (URANS) approach with a physical model of cavitation [19,37]. While the cavitation effects of hydrofoils have been well-investigated, studies of ventilation are still scarce [14,38]. As ventilation is a highly complex phenomenon for which systematic data are very difficult to obtain either experimentally or numerically, it is not explicitly considered in the preliminary design phase discussed in this paper.

Cavitation is an unsteady phenomenon that takes place in water when the local pressure on a surface is below the saturated vapor pressure of water, with bubble and vapor formation taking place along the lifting surface. This is known to cause "cavitation damage" in the form of corrosion of rotor blades of boat motors. In the case of aircraft, a disadvantageous effect is the increase in drag of the aircraft during takeoff and possibly some form of cavitation damage along the hull for certain configurations. Cavitation performance is usually compared via the non-dimensional cavitation number, defined as:

$$Ca \equiv \frac{p_\infty - p_V}{\frac{1}{2}\rho_W u_\infty^2}, \quad p_\infty = p_{\text{atm}} + \rho_W gh, \tag{4}$$

where $p_\infty$ is the freestream pressure, $u_\infty$ is the freestream speed, $p_{\text{atm}}$ is the atmospheric pressure, $p_V$ is the vapor pressure, and $h$ is the reference height corresponding to the hydrostatic pressure head.

The condition for cavitation inception over hydrofoils is given as $-C_{p_{\min}} \geq Ca$, where $C_{p_{\min}}$ is the minimum pressure coefficient over the hydrofoil. As a result, cavitation inception usually occurs on the upper surface of foils as the local pressure is lower than the freestream pressure for foils under lifting conditions. Numerical studies in hydrodynamic analysis and optimization for foil shapes usually consider variations of lift and drag forces against cavitation numbers.

Another issue involving cavitation inception is that the Reynolds, Mach, and cavitation numbers now play roles in the similarity analysis of flows over hydrofoils due to the flow's additional dependence on the vapor pressure of the fluid in liquid state. If a particular Reynolds number is set for analysis, its cavitation and Mach numbers are determined for a fixed reference length. As the cavitation number is independent of a length scale, adjusting the reference length to obtain a freestream speed for a fixed Reynolds number adjusts the cavitation number accordingly. This changes the characteristics of cavitation formation over the hydrofoil, as the pressure coefficient conditions for cavitation inception change, viz. the location of incipient cavitation over the hydrofoil surface will change.

The major issue with existing studies of hydrofoils is that they have originally been designed for cruising speeds of ships, which correspond to certain cavitation numbers as their design points. In the water-takeoff regime for an amphibious aircraft, such a fixed design point does not apply as the flow is accelerating.

## 3. Research Methodology

The work presented in this paper focuses on the hydrofoil design and analysis. The general preliminary design, weight estimation, and sizing procedures for the aircraft are therefore outside the scope of this work. For the discussion presented below, we

assume that the aforementioned procedures have been completed and the maximum takeoff weight, the required wing area, aircraft stall speed, and the powerplant selection have been determined. A hull selection and design are also already determined based on hull resistance and trim angle variations with speed. The Douglas Sea Scale [39] is used to measure the height and swell of a sea on a 0–10 degree scale. In this paper, a level 0 (no wave) condition in calm water is assumed in all considerations for the initial design. The high-fidelity computational analyses are performed in 2D with a fixed reference length based on the sizing procedure. The 2D analyses are deemed suitable for the preliminary design stage [29]. Performing high-fidelity 3D analyses that involve multiphase flows could be computationally prohibitive and would be more appropriate at detailed design stages.

The drag force caused by the empennage is considered as negligible in all equations of motion compared to the forces generated by the wing and hydrofoil. All aerodynamic forces are assumed to be concentrated as point loads from the aerodynamic center of the respective lifting surface, assumed to be located at 25% of the chord length from the leading edge. No control surfaces for the hydrofoils will be considered in the current investigation as the design framework considers the stability without the need of these devices.

Based on these assumptions for the design scope, the hydrofoil conceptual design and preliminary sizing, water-takeoff analysis, and stability analysis are discussed in this section.

### 3.1. Hydrofoil Conceptual Design Framework and Preliminary Sizing

The following design procedure considers hull-based amphibious aircraft instead of float-based, as the primary goal is to minimize large hull contact with water. This section provides guidelines to determine the suitable hydrofoil configuration, geometry, and location for preliminary design purposes.

#### 3.1.1. Configuration and Profile Selection

There are many possible hydrofoil configurations for amphibious aircraft including, but not limited to, surface-piercing hydrofoils, foldable-protruding hydrofoils, and strut-based hydrofoils [3,24]. As the name suggests, a surface-piercing hydrofoil has a hydrofoil protruding directly from the hull of the aircraft that pierces the water surface, such as the ones in LISA Akoya. When such a hydrofoil is foldable, it is considered as a foldable-protruding hydrofoil. The last category refers to a hydrofoil with a strut-based structure extending from the body of the aircraft. The surface-piercing and strut-based hydrofoil configurations are depicted in Figure 1. For the water-takeoff consideration in our design and analysis discussion, the hydrofoils are assumed to be fully deployed. The major dimensional constraints that will be discussed are relevant to this state, as well.

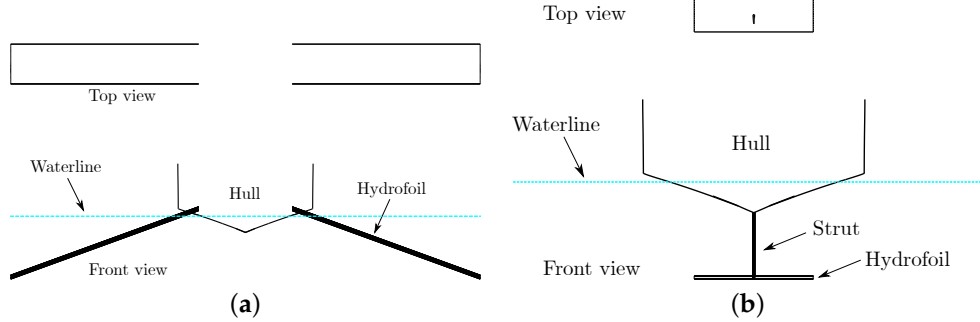

**Figure 1.** Hydrofoil configurations. (**a**) Surface-piercing Hydrofoil; (**b**) Strut-based Hydrofoil.

The number of hydrofoils on the aircraft is an important consideration in the weight and stability estimation of the initial design. The LISA Akoya implements two hydrofoils, one set near its wing and one set below the empennage; this configuration seems to eliminate the complex hull design considerations, such as stepped and planing configura-

tions, and enables the design of more aerodynamic fuselages as the hydrofoils provide lift throughout the water-takeoff run and reduce the wetted hull area.

To select an appropriate hydrofoil profile, we identify several criteria to ensure efficient performance. These criteria include: low hydrofoil drag coefficient within the entire takeoff speed range, and high lift-to-drag ratios within the speed range in which hull resistance is dominant. To ensure stability, the rate of change of the hydrofoil moment coefficient with respect to angle of attack should be negative within the range of hull trim angle. A higher hydrofoil lift coefficient is desirable as it will reduce the required hydrofoil area, to be further discussed in Section 3.1.2.

There are three main classifications for cavitation of hydrofoils, namely (1) subcavitating, where flow is fully attached over the hydrofoil; (2) partially cavitating, where flow separation of water takes place at some transition point with vapor cavities partially forming over the upper surface of the hydrofoil; and (3) supercavitating, where the hydrofoil undergoes separated flow from water with the formation of a large vapor cavity over the entire upper surface [3]. Past literature has shown hydrofoils have been designed to be efficient in either subcavitating or supercavitating regimes. Note that the assumption of the "aerodynamic center" of the hydrofoil located at approximately 25% of the chord length does not apply for these foils under cavitating conditions. The available databases of hydrofoils (e.g., University of Illinois at Urbana-Champaign (UIUC) Airfoil Database: https://m-selig.ae.illinois.edu/ads/coord_database.html) pale in comparison to their aerodynamic counterparts and a designer must resort to sifting through historical literature to find shapes. Moreover, the cavitation phenomenon will change the flow performance characteristics in hydrofoils compared to those of airfoils, as will be seen in the lift and drag coefficient profiles presented in Section 4.

The estimated range for the minimum pressure coefficient to ensure a subcavitating regime for an amphibious aircraft can be determined by its design water-takeoff speed. Consider, for instance, the LISA Akoya, a small two-seater aircraft, which has a stall speed of $u_s$ = 23 m/s and an estimated average hydrofoil height of 1 m from the free surface (http://lisa-airplanes.com/en/light-amphibious-aircraft-akoya/equipment-performance/). Estimating its takeoff speed at sea level by $u_{TO}$ = 1.2, $u_s$ = 27.6 m/s, its cavitation number constraint is the following:

$$ -C_{p_{\min}} < \frac{101,325 + 998.2 \times 9.81 \times 1 - 3640}{\frac{1}{2} \times 998.2 \times 27.6^2} \approx 0.2827. \tag{5} $$

This means that the lowest pressure coefficient over the hydrofoil must be $C_{p_{\min}} \geq -0.2827$ to prevent incipient cavitation; it is unlikely for subcavitating foils to satisfy this pressure coefficient constraint with beneficial $(L/D)$ ratios. Thus, at the required speeds for water-takeoff, which are bound to be higher for larger aircraft, cavitation is almost an inevitable phenomenon as the cavitation number reduces with increase in dynamic pressure underwater, indicating that supercavitating hydrofoils are the only viable solution. Thurston and Vagianos showed that hydrofoils designed for supercavitating regimes far outperformed their subcavitating counterparts in terms of $(L/D)$ ratios as a result of negligible effects of ventilation in supercavitating conditions, and should be primary considerations for amphibious aircraft [3]. A study on the performance of hydrofoils designed for subcavitating and partially cavitating regimes for use in amphibians supports this conclusion [1]. It is thus imperative to use supercavitating hydrofoils in amphibious aircraft applications.

Experimental data exist for some supercavitating profiles [40], which can be used as a reference when sizing the hydrofoil. The attainable maximum lift coefficient for supercavitating hydrofoils appears to be $C_{L_{hf_{max}}} \approx 0.27$ including ventilation effects in the supercavitating regime according to Thurston's review in the 1970s [2]. However, some supercavitating profiles, such as the SCSB profiles developed by Brizzolara [14], have shown improved performance in subcavitating and partially cavitating conditions in terms of $(L/D)_{hf}$ ratios compared to previous designs.

### 3.1.2. Preliminary Sizing

The design considerations and criteria to determine hydrofoil geometry parameters including area, sweep angle, taper, dihedral, and aspect ratio are summarized and presented here. The selection is mainly based on the physical properties and hydrofoil performance. The span and incidence angle selection process, which relies on a water-takeoff minimization procedure, will be described separately in Section 3.1.3.

#### 3.1.2.1. Hydrofoil Area

The hydrofoil needs to generate lift to unport the hull from the water while it is submerged. The stall speed of the hydrofoil ($u_{hf_s}$) is defined as the lift required by the hydrofoil to counter the weight of the aircraft minus the lift of the wing at the same speed. The required area for this condition can be derived from the results of the preliminary wing and horizontal tail sizing procedures by equating this stall speed to the speed at which the horizontal tail's elevator ($u_{h_e}$) is able to provide effective corrections as an initial design point.

$$u_{hf_s} = u_{h_e}. \tag{6}$$

The required area $S_{hf_{\text{req}}}$ is then determined by equating the lift generated by the hydrofoil at its stall speed to the difference between the weight of the aircraft $W$ and the lift generated by the wing in the same condition:

$$\frac{1}{2}\rho_W u_{hf_s}^2 S_{hf_{\text{req}}} C_{L_{hf\text{max}}} = W - \frac{1}{2}\rho_A u_{h_e}^2 S_w C_{L_w} \tag{7}$$

$$\implies S_{hf_{\text{req}}} = \frac{2W - \rho_A u_{h_e}^2 S_w C_{L_w}}{\rho_W u_{hf_s}^2 C_{L_{hf\text{max}}}}, \tag{8}$$

where $W$ is the aircraft weight. A high $C_{L_{hf\text{max}}}$ will therefore reduce the effective hydrofoil area, which reduces the drag generated by the hydrofoil. Note that this is the required area for a hydrofoil with no dihedral to generate sufficient lift for the aircraft weight, which can be slightly generalized as will be discussed in the dihedral sizing description below.

The determination of $C_{L_w}$ depends on the Reynolds number, set by $u_{h_e}$, and $\alpha_w$, determined by the wing incidence angle $\alpha_{w_i}$ and hull trim angle at the corresponding point during the water-takeoff. As a first estimation in the design stage, $C_{L_w}$ can be determined via quick, low-fidelity analysis tools, such as XFOIL [41]. The value of $C_{L_{hf\text{max}}}$ depends on the hydrofoil profile selection, as mentioned in Section 3.1.1. This method implicitly accounts for the losses of lift due to ventilation effects as a heuristic, assuming that the hydrofoil will not yield the ideal maximum lift coefficient at this design point due to ventilation.

#### 3.1.2.2. Sweep Angle and Taper

King and Land [42] observed that the implementation of sweep in subcavitating conditions delayed the onset of cavitation inception to higher speeds for angles between 0–45° at the cost of decreased $(L/D)$ ratios, so its implementation might be beneficial for aircraft designed for water-takeoffs in only subcavitating conditions for hydrofoils. It is, however, not efficient for supercavitating hydrofoils as they are designed for optimal performance in supercavitating conditions. They also observed that the implementation of taper has little effect on the characteristics of the flow and the performance.

#### 3.1.2.3. Dihedral Angle

For surface-piercing hydrofoils, anhedral would be required to maintain lifting performance while the hull is unported from the water. The effective area of the hydrofoil with

a dihedral angle $\delta_{hf}$ is then given by $S_{hf} = S_{hf_{\text{req}}} / \cos^2 \delta_{hf}$, which is then substituted into Equation (8) to give:

$$S_{hf} = \frac{2W - \rho_A u_{h_e}^2 S_w C_{L_w}}{\rho_W u_{hf_s}^2 C_{L_{hf\max}} \cos^2 \delta_{hf}}. \tag{9}$$

For a hull-protruding hydrofoil with a generic foil profile to be a beneficial lifting surface while reducing hull contact area, $\delta_{hf}$ should be between $-45°$ and $0°$ to enable unporting the hull while still being submerged, as the hydrofoil should protrude below the base of the hull and not above.

### 3.1.2.4. Aspect Ratio

The aspect ratio ($AR$) must be set, for a hull-protruding hydrofoil with positive $\delta_{hf}$, such that the height of the deployed hydrofoil is not greater than the distance of the landing gear's wheels from the aircraft's center of gravity (CG, $z_{lg}$).

$$AR_{hf} \leq \frac{a z_{lg}^2}{S_{hf_{\text{req}}} \tan^2 \delta_{hf}}, \quad 0 < a < 1, \tag{10}$$

where $a$ is a factor of safety that can be set by the designer to account for ground strikes during takeoffs on land in case of mechanical failures.

The aspect ratio selected should ideally be high (i.e., greater than 5) according to Thurston [3]. For aircraft with strut-based hydrofoils, this may result in the effective span of the hydrofoil being larger than the fuselage width. In these cases, retractable/folding mechanisms should be considered in the design of the strut to fold the hydrofoil and retract it into the fuselage/hull. The determination of the aspect ratio is further investigated by optimization of the span length of the hydrofoil, discussed in Section 3.1.3.

### 3.1.3. Span and Incidence Angle Optimization

In this work, the hydrofoil's span and incidence angle are determined via the minimization of the water-takeoff distance, in which they are treated as design variables. The water-takeoff distance calculation, as the objective function, will be elaborated in Section 3.2. A lift constraint is considered, to model that the total lift provided by the lifting surfaces (wing, hydrofoil and horizontal tail) does not exceed the aircraft weight during the water-takeoff procedure, to satisfy the inequality constraint in residual form $\mathcal{R}(L_{hf}) = L_{hf} + L_w + L_h - W \leq 0$. The variable bounds for the speed and angle of attack, which are used in the water-takeoff calculation, also need to be specified. The design limit of aspect ratio, which is related to the span as $b_{hf} = \sqrt{AR_{hf} \times S_{hf}}$, is also imposed following the discussion in Section 3.1.2. The optimization procedure is summarized in Table 1.

**Table 1.** Optimization problem formulation for the optimal hydrofoil span and incidence angle.

| Optimization | Function Variables | Description |
|---|---|---|
| Minimize | $x_W$ | Water-takeoff distance. |
| Design variables | $b_{hf}$ $\alpha_{hf_i}$ | Span length of the hydrofoil. Incidence angle of the hydrofoil. |
| Constraints | $L_{hf} + L_w + L_h \leq W$ | The total lift provided by the lifting surfaces must not exceed the aircraft weight. |
| Bounds | $0 < u \leq u_{TO}$ $\alpha_{\min} \leq \alpha_{hf} \leq \alpha_{\max}$ $AR_{hf_{\min}} \leq AR_{hf} \leq AR_{hf_{\max}}$ | Speed range within takeoff regime. Angle of attack within operational bounds. Aspect ratio within design bounds. |

The corresponding extended design structure matrix (XDSM) (https://github.com/mdolab/pyXDSM) representation [43] is shown in Figure 2. In this representation, mathematical functions with well-defined inputs and outputs are colored in green, implicit components that solve residual equations are colored in red, optimizations and design of experiments are colored in blue, solvers pertaining to physics are colored in orange, and metamodels are colored in yellow. Data links between components are presented as thick, gray lines with slanted boxes denoting the communicated variables; inputs into components are fed "right-down" and outputs from components are returned "left-up", depending on the order of execution. Inputs and outputs that are not connected to any other systems, such as initial guesses (denoted with $-_0$ subscripts) and optimization results (denoted with $-^*$ superscripts) of some optimizations, are shown as parameters inside white boxes. Processes are indicated with black lines representing arrows.

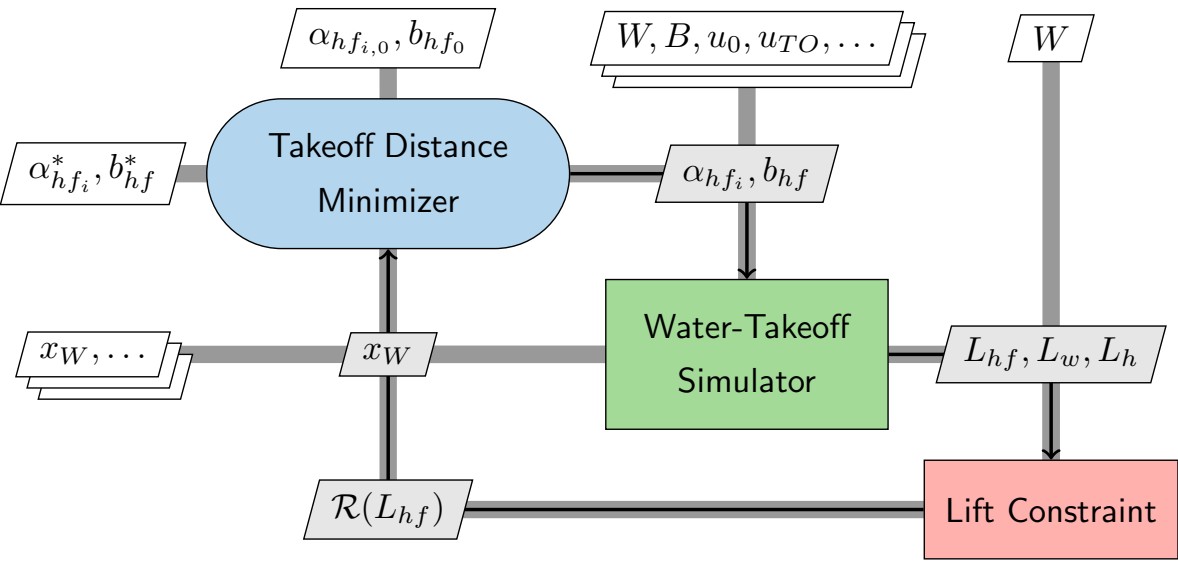

**Figure 2.** XDSM diagram for the hydrofoil span and incidence angle optimization.

In this study, we only consider hydrofoils with rectangular planforms, as sweep and taper have no appreciable effects on the hydrofoil performance from the discussion in Section 3.1.2.2. To start the optimization procedure, an initial span length is selected based on the initial guess of the aspect ratio. This span length is then used to determine the chord length $\bar{c}_{hf}$ of the hydrofoil that is used in computational fluid dynamics (CFD) analyses, elaborated in Section 3.4.2.

The hydrofoil must be able to operate at the optimum angle for efficient performance in the takeoff regime. The angle of attack of the aircraft is subject to changes during water-takeoff due to hull designs and wave formations, which results in changes of the angle of attack of the hydrofoil. Hence, the incidence angle should be determined by accounting for the hull trim angle while considering the required angle of attack for the chosen hydrofoil profile:

$$\alpha_{hf_i} = \alpha_{\text{trim}} - \alpha_{hf}. \tag{11}$$

The angle of attack varies with time as a result of the hull's dynamic trim angle. The consideration of an actuated system for adjusting the angle of attack, while possible, is too complex for an initial design consideration and is deemed beyond the scope of the present study. Note that such an implementation would add structural weight, which needs to be considered in the design process. The lift constraint is considered for the reason that if the aircraft and the hydrofoil are unported from water, the loss of lift from the hydrofoil

is extremely large due to the change in density of the fluid from water to air. This would result in the weight exceeding the total lift in this condition, and the aircraft would crash onto the water, resulting in unstable effects on the resistance and trim.

### 3.1.4. Location Optimization

The hydrofoil location in the amphibious aircraft is determined primarily based on the stability and trim requirements. The hydrofoil must generate minimal moments about the aircraft center of gravity (CG) until the speed at which the elevator is effective enough to provide corrections is reached, while simultaneously providing optimal lift to minimize hull resistance. This objective is attained by considering minimization of the maximum stabilizer force required during a simulated water-takeoff run. The reasoning for the consideration of this objective function will be presented in Section 3.3, and the water-takeoff simulation model will be described in Section 3.2. The optimization problem formulation to determine the hydrofoil location is summarized in Table 2. The hydrofoil location is indicated by the coordinates of its reference point, $(x_{hf}, z_{hf})$ with respect to the CG as origin. The optimum span and incidence angle from the takeoff distance optimization problem in Section 3.1.3 are provided to the stabilizer force minimization problem, which determines the optimal hydrofoil position. The XDSM diagram is presented in Figure 3.

**Table 2.** Optimization problem formulation to determine the hydrofoil's location.

| Optimization | Function Variables | Description |
|---|---|---|
| Minimize | $\max|L_h|$ | Maximum horizontal stabilizer force required during takeoff. |
| Design variables | $(x_{hf}, z_{hf})$ | Coordinates of hydrofoil's reference point. |
| Bounds | $x_{\text{nose}} \leq x_{hf} \leq 0$ <br> $z_{lg} \leq z_{hf} \leq z_{\text{base}}$ | Horizontal position between aircraft nose and CG. <br> Vertical position between landing gear and aircraft base. |

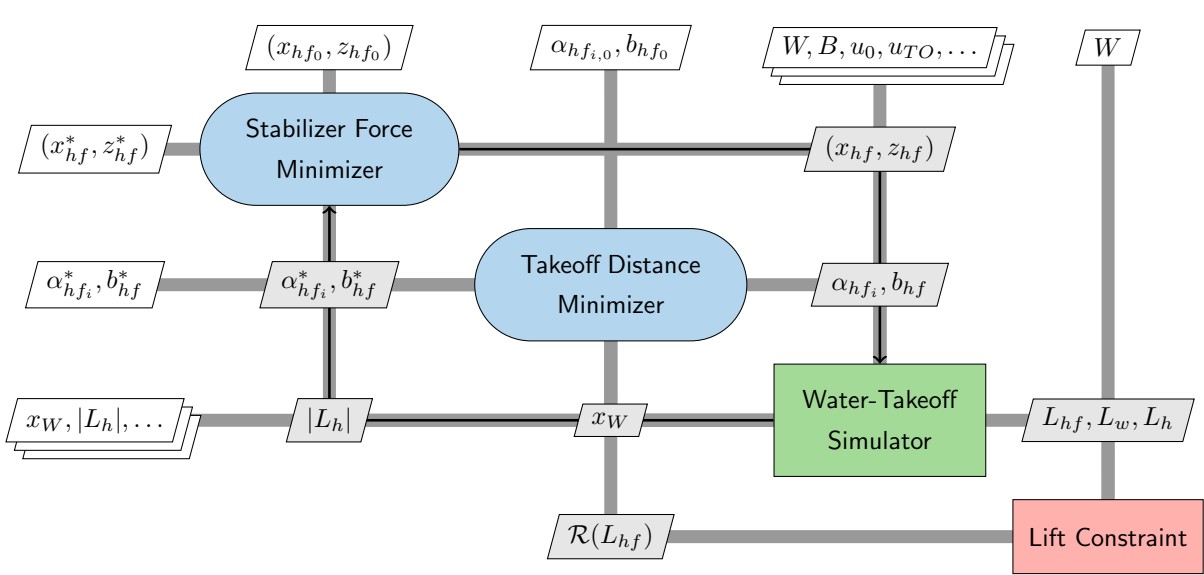

**Figure 3.** XDSM diagram for the hydrofoil location optimization.

### 3.2. Water-Takeoff Analysis

In this analysis, the water-takeoff distance is defined as the horizontal distance traveled by the aircraft during acceleration in water, to reach a pre-determined speed at which the

aircraft generates sufficient lift to achieve takeoff at a particular angle of attack. The water-takeoff analysis is performed based on Newton's second law, as follows:

$$m\dot{u} = \underbrace{T\cos\phi_t,}_{\text{Thrust}} \tag{12a}$$

$$-\underbrace{\rho_W g B^3 \frac{C_\Delta}{C_{\Delta_0}} C_R,}_{\text{Wave resistance}} \quad \text{where} \quad \frac{C_\Delta}{C_{\Delta_0}} = 1 - \frac{\rho_W g B C_V^2 C_{L_w}}{2(W/S_w)} - \frac{C_{L_{\text{hull}}}}{C_{\Delta_0}}, \tag{12b}$$

$$-\underbrace{\frac{1}{2}\rho_W S_{\text{wet}} u^2 \frac{C_\Delta}{C_{\Delta_0}} C_F,}_{\text{Viscous resistance}} \tag{12c}$$

$$-\underbrace{\frac{u^2}{2}\left[\rho_A\left(C_{D_w}\cos\alpha_w + C_{L_w}\sin\alpha_w\right) + \rho_W\left(C_{D_{hf}}\cos\alpha_{hf} + C_{L_{hf}}\sin\alpha_{hf}\right)\right],}_{\text{Contributions from wing and hydrofoil lift and drag}} \tag{12d}$$

where $m$ and $\dot{u}$ denote mass and acceleration, respectively. Equation (12d) consists of the contributions towards lift and drag from the wing and hydrofoil at their respective angles of attack, and Equation (12a) is the thrust force $T$ at some thrust inclination angle $\phi_t$. The remaining force terms on the right-hand side, such as the hull resistance and viscous resistance, are described in detail below.

Equation (12b) represents the wave resistance (unrelated to wave drag) of the hull with respect to variations in its trim angle based on towing tank tests. This term is scaled by the submerged volume ratio $C_\Delta/C_{\Delta_0}$ at a given instant during takeoff with respect to the submerged volume at rest. At each point of the water-takeoff calculation, the 'effective weight' of the aircraft (its weight minus the total lift) is considered to be balanced by the buoyant forces, hence $\Delta = W - \sum_i L_i$, $i \in \{hf, h, w\}$, which is non-dimensionalized to give $C_\Delta$. During the planing stage, the hull contact with water is minimal, i.e., $C_\Delta/C_{\Delta_0} \approx 0$, where $C_{\Delta_0}$ is the maximum load coefficient in the absence of lift, but it may generate viscous skin-friction resistance proportional to a small wetted area $S_{\text{wet}}$ in contact with the water, hence an additional viscous resistance term is added for the planing stage given by Equation (12c). Determining this wetted area is complex without knowledge of the exact waterline height and the spray pattern, so an initial value is set based on a geometric estimation and then scaled by the submerged volume ratio $C_\Delta/C_{\Delta_0}$. The skin-frictional resistance coefficient $C_F$ is obtained via the ITTC-57 formula, based on data from empirical towing tank tests [31]:

$$C_F = \frac{0.075}{\left(\log_{10} Re - 2\right)^2}. \tag{13}$$

During a water-takeoff run for an amphibious aircraft without a hydrofoil, this viscous resistance term dominates the drag in water during the planing stage. In the model considered, when the hull is unported from the water with the help of the hydrofoil, viz. the buoyant force is zero, this term is also zero.

The kinematic equations are modeled via a time-stepping approach, in which the speed $u$, time $t$, and displacement $x$ variables are discretized, with subscripts $0 \leq i \leq N_t$ for $N_t$ timesteps. The dot notation is used to denote time derivatives. This approach is similar to the water-takeoff model for amphibious aircraft used in the work of Qiu and Song [8]; they did not, however, consider the implementation of a hydrofoil in their study. The following equations are solved for the speed and displacement by integration of the equations of motion for some initial condition ($t_0$, $x_0$, $u_0$, $\dot{u}_0$). The output consists of

variables including the water-takeoff distance $x_W$, the time taken $t_W$, and the other relevant parameters for analysis and optimization.

$$u_i = u_{i-1} + \dot{u}_i(t_i - t_{i-1}), \tag{14a}$$

$$x_i = x_{i-1} + \frac{1}{2}(u_i + u_{i-1})(t - t_{i-1}). \tag{14b}$$

In the water-takeoff regime, the wetted area of the hydrofoil underwater with dihedral angle $\delta_{hf}$ (negative, hence anhedral) reduces as the waterline height of the aircraft decreases due to the lift generated by the hydrofoil. The lift generated by the submerged area of the hydrofoil with cavitation effects as an initial approximation can be modeled as:

$$L_{hf} = \frac{1}{2}\rho_W u^2 C_{L_{hf}} S_{hf_{\text{wet}}}(h), \tag{15}$$

where the submerged area $S_{hf_{\text{wet}}}(h)$, now a function of the height $h$, is assumed to be fully wetted underwater, and the lift coefficient of the hydrofoil takes the losses due to cavitation effects into account. For an initial approximation assuming the waterline interface as a flat surface, this underwater wetted area can be determined geometrically as:

$$S_{hf_{\text{wet}}}(h) \approx \frac{S_{hf_{\text{req}}}}{b_{hf}^2}\left(b_{hf} + \frac{h}{\sin \delta_{hf}}\right)^2, \quad -90° \leq \delta_{hf} < 0°. \tag{16}$$

To obtain the lift and drag forces over the hydrofoil, either experiments or simulations including at least cavitation modeling need to be performed to obtain a reasonable estimate at this stage of the design process. It is computationally very expensive to determine the hydrodynamic forces via high-fidelity CFD at every time-discretized point in a takeoff analysis as observed by Seth and Liem [44], so a surrogate model, or an approximation model, is generated for use in the water-takeoff analysis for the optimization problems to reduce the computational burden. The surrogate models will compute the non-dimensional coefficients of the hydrofoil as functions of speed and angle of attack.

In this work, sample-based surrogate models are selected for their simplicity and non-intrusive nature. To construct the surrogate models, we first need to generate samples by running the high-fidelity analyses. A design of experiments is performed to generate the design space by selecting $N$ in the speed range $u_0 \leq u \leq u_{TO}$, with initial speed $u_0$, to generate a set $\{u\}$ and $N_\alpha$ samples in the angle of attack range $\alpha_{\min} \leq \alpha \leq \alpha_{\max}$ to generate a set $\{\alpha\}$. The Cartesian product of these sets defines the design space to generate CFD training and testing data for the surrogate, as will be discussed in Section 4.2.2. Similar arguments can be applied to airfoils, so surrogate models are also considered for them in the framework, although they may not be necessary with the use of low-fidelity solvers, which is to be further discussed in Section 3.4.2. The water-takeoff analysis with surrogate models is presented as an XDSM in Figure 4.

Note that this takeoff procedure alone does not guarantee the aircraft will satisfy the required stability and trim characteristics, which are considered in the following section to further elaborate the considerations for the optimization problem presented in Table 2.

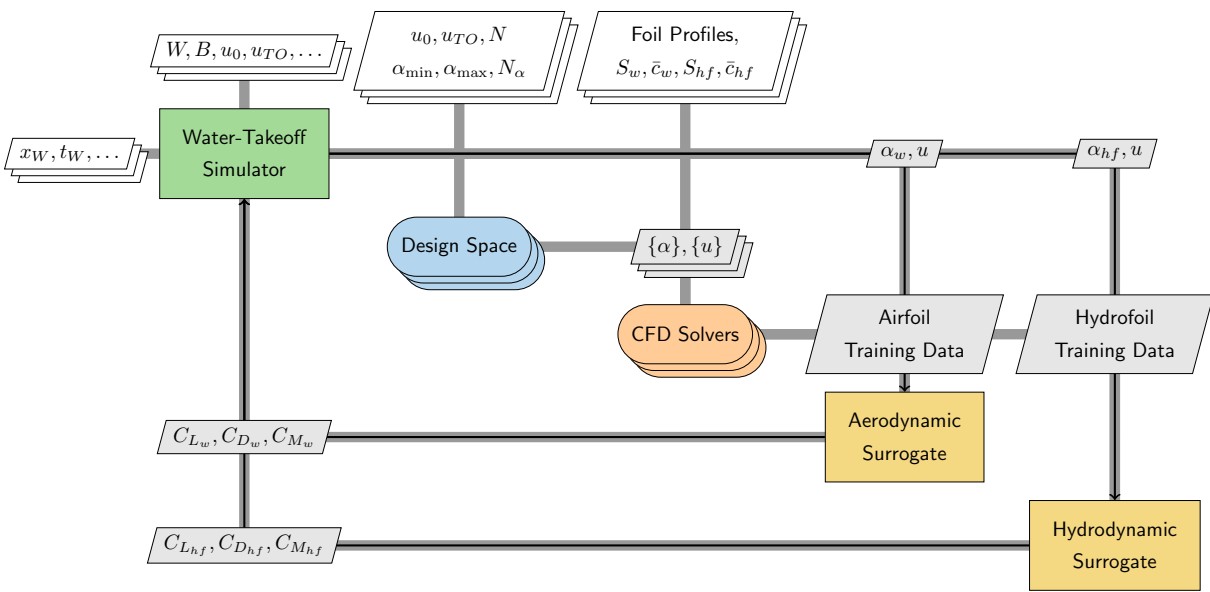

**Figure 4.** XDSM diagram for the water-takeoff simulator with surrogate models.

### 3.3. Stability Considerations and Analysis

In this section, we first discuss the physical considerations in evaluating the longitudinal stability requirements of amphibious aircraft, followed by the details of trim analysis. As we assume a no-wave condition, the lateral stability conditions caused by irregular wave patterns and effects of sponsons during water-takeoff are not considered at this preliminary design stage.

#### 3.3.1. Physical Considerations

Consider Cartesian coordinates indicating the position of the aerodynamic center of the hydrofoil $(x_{hf}, z_{hf})$ where the origin is set at the aircraft CG, assumed to be located at 33% MAC of the wing aft of the leading edge for the following analysis. The large lift and drag forces produced by the hydrofoil will generate significant moments. Looking at an amphibious aircraft with its nose facing left, the hydrofoil will be located below the CG (thus, $z_{hf} < 0$), and its drag will resultantly contribute a nose-down pitching moment, which must be corrected by the elevator to maintain the required hull trim angle. To minimize elevator effort, it is best to counter this anti-clockwise moment by placing the hydrofoil ahead of the CG (thus, $x_{hf} < 0$), so the hydrofoil is able to provide lift that also provides an opposing pitch-up moment to compensate for the moment caused by its drag. This motivates a positioning procedure, such that the minimum takeoff distance is achieved during the water-takeoff run while maximizing moments generated by lift and minimizing moments generated by drag. A complication of this procedure is that the lift and drag of the hydrofoil suffer from variations in speed and angles of attack dissimilar to their airfoil counterparts due to cavitation and ventilation effects and are, thus, not predictable via tools meant for airfoil analysis; hence, the considerations using high-fidelity CFD and surrogate models introduced in Section 3.2. The effects of ventilation, however, are not explicitly modeled in this work, for reasons described in Section 2.2.

#### 3.3.2. Trim Analysis

The non-dimensionalized moment equation is derived in the body-axis system of the aircraft following the process of Raymer [45] with modifications for the hydrofoil. The drag of the hydrofoil is not considered as negligible in this derivation unlike as usually considered for the tail, as the density of water is approximately three orders of magnitude

greater than that of air. The following expression is solved for the trim condition of the moment coefficient about the center of gravity $C_{M_{cg}} = 0$, to determine the required $C_{L_h}$ to correct the moments generated by the hydrofoil and the wing.

$$C_{M_{cg}} = C_{M_w} + C_{M_{eng}} + C_{L_w}(\bar{x}_{cg} - \bar{x}_{ac_w}) - \eta_h V_h C_{L_h} + \eta_{hf}\left[V_{hf}C_{L_{hf}} - Z_{hf}C_{D_{hf}}\right], \quad (17)$$

where $C_{M_{eng}}$ is the moment coefficient corresponding to the engine, $\bar{x}_{cg} = x_{cg}/\bar{c}_w$ is the non-dimensionalized location of the center of gravity and $\bar{x}_{ac_w} = x_{ac_w}/\bar{c}_w$ is the location of the aerodynamic center of the wing with respect to the nose of the aircraft as the origin. The dynamic pressure ratios $\eta_h$ and $\eta_{hf}$ are defined as:

$$\eta_h \equiv \frac{\frac{1}{2}\rho_A u_h^2}{\frac{1}{2}\rho_A u_w^2}, \quad \eta_{hf} \equiv \frac{\frac{1}{2}\rho_W u_{hf}^2}{\frac{1}{2}\rho_A u_w^2}. \quad (18)$$

The longitudinal stability coefficients for a component $c$ corresponding to a lifting surface are defined as:

$$V_c \equiv \frac{S_c}{S_w}\frac{l_c}{\bar{c}_w}, \quad Z_c \equiv \frac{S_c}{S_w}\frac{z_c}{\bar{c}_w}, \quad (19)$$

with moment arms $l_{hf} \equiv x_{cg} - x_{hf}$, $z_{hf} \equiv z_{cg} - z_{hf}$, which will be determined via the optimization procedure for the hydrofoil location presented in Section 3.1.4.

### 3.4. Design Framework and Solvers

In this section, we summarize the entire framework of the hydrofoil design process described in the previous sections, and present the relevant computational solvers for analyses.

#### 3.4.1. Overall Design Process

The entire design process is depicted in the XDSM presented in Figure 5, with the descriptions of diagram components are as described in Section 3.1.3. The water-takeoff simulator is first executed without any hydrofoil to obtain the initial aircraft design's water-takeoff performance. The hydrofoil design parameters, determined from the preliminary sizing, are then used for the generation of its surrogate model using high-fidelity CFD, and similarly for the wing if required, for the design space of takeoff speeds and operational angles of attack. These surrogate models are used in the minimization of the water-takeoff distance to determine the optimal span and incidence angle of the hydrofoil. These optimal values are then provided to the optimization loop to minimize the required stabilizer force to achieve trim during the water-takeoff run by adjusting the position of the hydrofoil for the optimum. The relevant data, such as the optimized takeoff distance, span length, incidence angle, minimum stabilizer force, hydrofoil location, and the data obtained from the water-takeoff procedure, are provided to the designer to assess the performance and feasibility of the design.

#### 3.4.2. Computational Fluid Dynamics Solvers

The choice of the computational method of CFD for the determination of the coefficients of the airfoil and hydrofoil is extremely important, depending on the appropriate compromise between accuracy and speed at the required stage of the design process. Reasonably accurate low-fidelity solvers with good computational efficiency are openly available for airfoil analyses, but the same are not openly available for hydrofoil analyses with cavitation models, so open-source high-fidelity solvers are considered instead. In this work, low-fidelity solvers are deemed sufficient to evaluate the aerodynamic performance of aircraft wing within the operational regime considered in this study, while high-fidelity solvers are required to evaluate the hydrodynamic performance of hydrofoils, as described briefly below.

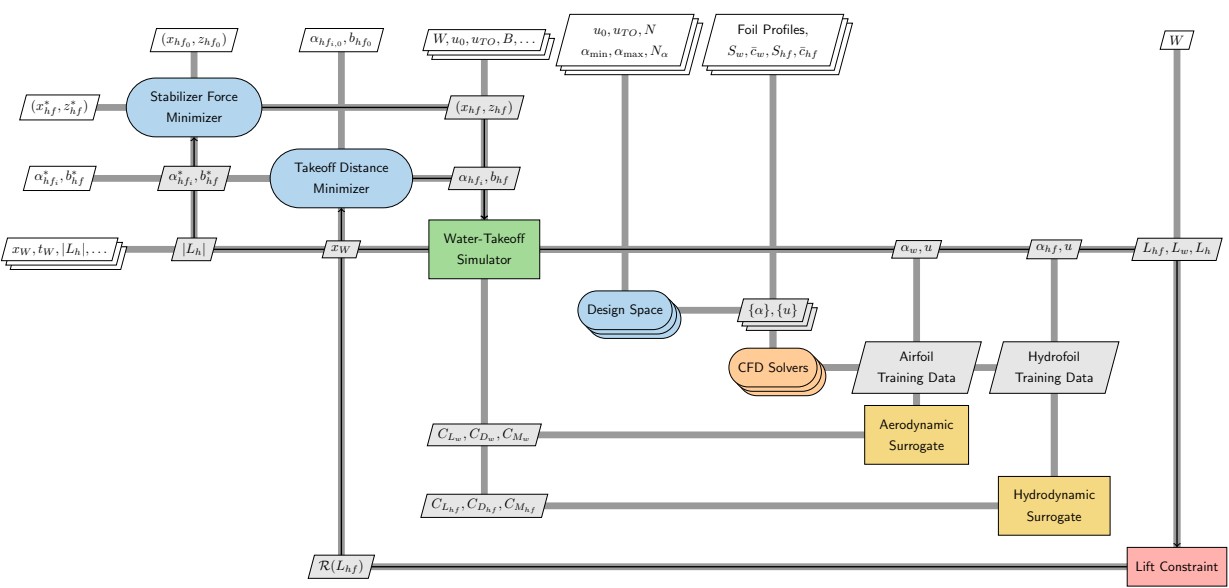

**Figure 5.** XDSM diagram for the hydrofoil design process.

### 3.4.2.1. Low-Fidelity Solvers

To reduce computational burden, XFOIL [41], a well-established viscous-inviscid design and analysis tool for airfoils at low Reynolds numbers, can be used to compute the aerodynamic coefficients for the wing by computing the lift coefficients of the airfoil $C_{L_a}$ with flap $\Delta C_{L_f}$ and ground effect $\Delta C_{L_g}$ corrections in the takeoff simulation itself:

$$C_{L_w} = C_{L_a} + \Delta C_{L_f} + \Delta C_{L_g}. \tag{20}$$

Note that XFOIL may not be able to predict the coefficients for flows that occur at high angles of attack beyond the stall angle, in which case high-fidelity CFD must be used; however, angles greater than the stall angle are not usually reached until the aircraft rotates after reaching the takeoff speed. A surrogate model via high-fidelity CFD can also be generated for the airfoil to account for ground effects more accurately during takeoff.

### 3.4.2.2. High-Fidelity Solvers

The unsteady analyses with cavitation modeling over the hydrofoil are performed by solving the URANS equations via the `interPhaseChangeFoam` module of the OpenFOAM framework [46], following the approach of Vernengo et al. [29]. One difference from this analysis is that the Reynolds numbers for these cases are in the turbulent regime; hence, the Spalart-Allmaras turbulence model is used based on the approach of Garg et al. [28]. The cavitation model used in this analysis is the Schnerr-Sauer model [47], and the equations are solved using the segregated PIMPLE approach.

## 4. Case Study Description and Results—10-Seater Amphibious Aircraft

A preliminary design of a 10-seater aircraft based on a DHC-6 Twin Otter is considered for the case study. A planing hull from the National Advisory Committee for Aeronautics (NACA) Technical Note TN-2481 report [49] is adopted for this design, where we derive a curve-fit from the available experimental data for the water-takeoff simulation purpose, as shown in Figure 6. The aircraft weight is approximately $W = 5620 \times 9.81$ N, its wing area is approximately $S_w = 39.2$ m$^2$ with an incidence angle of $\alpha_w = 0°$. The takeoff speed is selected to be 44 m/s, similar to the Twin Otter's. The selection of the airfoil for the wing is the NACA 63(4)–12 profile. A surrogate model for this airfoil using RANS without ground effects has been generated by Seth and Liem [1], which is used here.

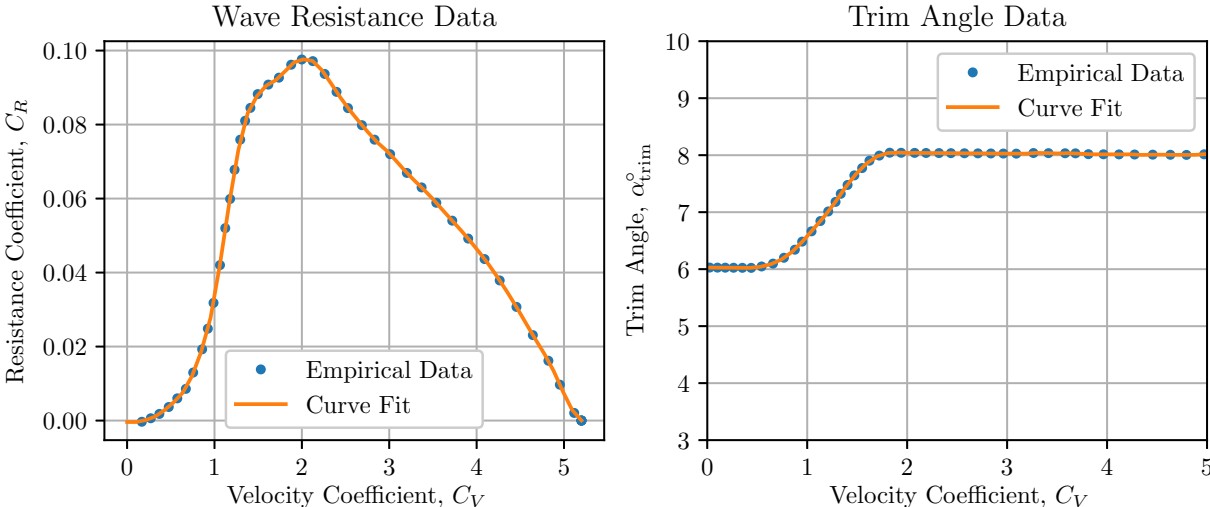

**Figure 6.** NACA TN-2481 planing hull data.

A quadratic thrust model presented by Gudmundsson [50] is used based on the specifications of a Pratt and Whitney Canada PT6A-34 turboprop engine via the following expression with $T$ as the thrust of the engine as a function of speed $u$, as shown in Equation (21). Figure 7 shows the thrust model only within the takeoff speed range of interest; note that the bounds of the domain make it look more linear than quadratic. The engine thrust is scaled for the first 10 s of the takeoff according to the expression in Equation (22) as a heuristic pilot model for the increase in thrust, similar to the case considered in Gudmundsson [50].

$$T(u) = \left( \frac{T_\text{static} - 2T_{u_\text{max}}}{u_\text{max}^2} \right) u^2 + \left( \frac{3T_{u_\text{max}} - 2T_\text{static}}{u_\text{max}} \right) u + T_\text{static}, \tag{21}$$

$$r(t) = \begin{cases} 0.25 + 0.75(t/10) & t \leq 10, \\ 0 & t > 10. \end{cases} \tag{22}$$

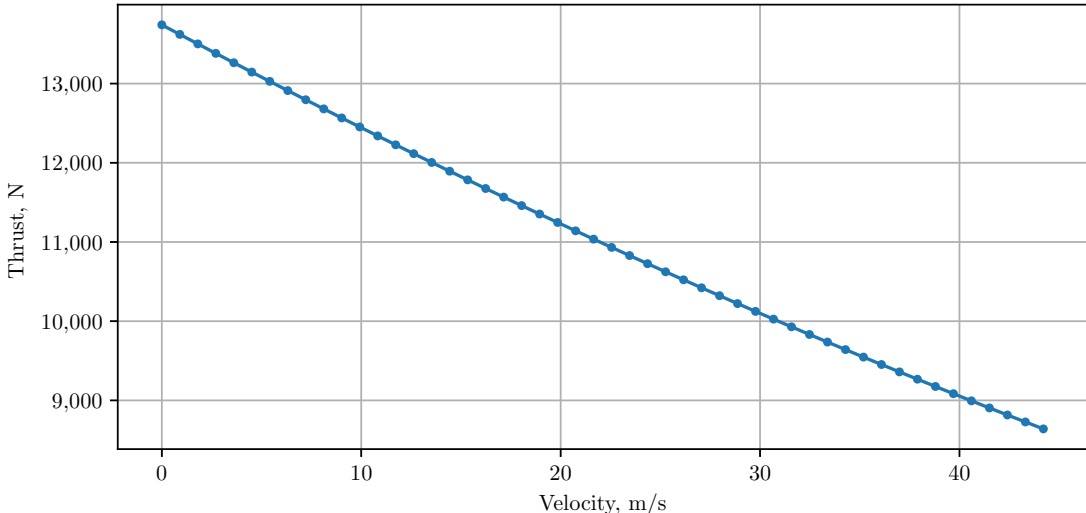

**Figure 7.** Thrust model as a function of speed, based on the specifications of a Pratt and Whitney Canada PT6A-34 turboprop engine.

### 4.1. Sizing and Profile Selection

The elevator effectiveness speed is considered to be $u_{h_e} \approx 15$ m/s. The wing lift at this speed with the corresponding trim angle at $\alpha_{\text{trim}} = 6°$ is estimated to be $C_{L_w} = 1.5$ using XFOIL and assuming $\Delta C_{L_f} + \Delta C_{L_g} = 0.3$. This does not account for interference effects between the wing and the fuselage. The hydrofoil profile selected for this case is the Waid-Lindberg hydrofoil profile [40] in a strut-based configuration. The cavitation number at this speed is $Ca \approx 0.87$, which is in the cavitating regime for the hydrofoil. The $C_{L_{hf_{\max}}}$ in this regime according to the experimental data is approximately 0.9, and the hydrofoil size can, hence, be determined by Equation (8), giving $S_{hf_{\text{req}}} \approx 0.455$ m$^2$. Considering a retractable, non-foldable hydrofoil system, an aspect ratio of 6 is selected initially, giving a chord length $\bar{c}_{hf} \approx 0.275$ m and initial span value $b_{hf_0} \approx 1.65$ m. The relevant parameters for the study are tabulated in Table 3.

**Table 3.** Aircraft and hydrofoil parameters.

| Parameter | Value |
|:---:|:---:|
| Wing Area, m$^2$ | 39.2 |
| Wing Aspect Ratio | 10.0 |
| Wing Span, m | 19.8 |
| Wing Chord, m | 1.98 |
| Wing Stall Speed, m/s | 36.8 |
| Wing Incidence Angle, ° | 0.0 |
| Wing Dihedral Angle, ° | 3.0 |
| Hull Beam Width, m | 1.71 |
| Takeoff Speed, m/s | 44.0 |
| Hydrofoil Area, m$^2$ | 0.454 |
| Hydrofoil Aspect Ratio | 6.0 |
| Hydrofoil Span, m | 1.65 |
| Hydrofoil Chord, m | 0.275 |
| Hydrofoil Mass, kg | 60.0 |

### 4.2. CFD and Surrogate Model Generation

This section discusses the use of CFD and surrogate models in the case study. The grid generation, CFD analyses, and surrogate model validations are described.

#### 4.2.1. Grid Generation, Convergence Study, and Validation

The meshes are generated using a hyperbolic grid generator via the Python module pyHyp (https://github.com/mdolab/pyhyp) developed by the Multdisciplinary Design Optimization Lab (MDOLab) at University of Michigan based on the theory from Luke et al. [51]. This generates structured hexahedral meshes which keep a low cell count with high quality, which are then converted into the unstructured OpenFOAM format for this study. The domain extends to approximately 30 chord lengths from the hydrofoil. The sharp leading edge of the hydrofoil is modified to create a small, blunt leading edge to ensure high quality grid generation, and a large number of elements is concentrated near the blunt trailing edge to resolve unsteady behaviour more accurately. At this preliminary design stage, we consider non-accelerating flows using a constant speed boundary condition.

The meshes and solver are validated using the experimental results from Waid and Lindberg [40] at $Re = 7.8 \times 10^5$, $Ca = 0.293$, $u = 9.144$ m/s, $\alpha_{hf} = 6°$, and three grids ($L_{0,1,2}$) are analyzed via Richardson extrapolation of the time-averaged drag coefficient $\overline{C}_{D_{hf}}$, using a grid refinement ratio of $\sqrt{N_{i-1}/N_i} = 2$, where $N_i$ is the number of cells of the coarsened grid with respect to the finer grid $N_{i-1}$. A wall distance of $2 \times 10^{-4}$ m is selected to match $y^+ > 30$ with the use of wall functions using the Spalart-Allmaras turbulence model to

avoid extremely small timesteps, which would drastically increase the computation time. The appropriate chord length $c_{hf}$ and non-dimensional time $\tau$ are selected such that the cavitating flow is developed over the entire chord length of the hydrofoil to determine the required simulation time $t$, given by Equation (23):

$$\tau = \frac{tu_{hf}}{\overline{c}_{hf}} \geq 4. \tag{23}$$

The $L_1$ grid and the leading edge modification mentioned previously are shown in Figure 8. The extrapolation is shown in Figure 9. The results are provided in Table 4, with a depiction of the cavity over the hydrofoil shown in Figure 10. Computing the relevant ratio of the grid convergence indices (GCI) for $\overline{C}_{D_{hf}}$ as the quantity of interest gives approximately $0.944 \approx 1$ within the asymptotic range and approximate solver convergence $p = 1.34$. The error from the $L_1$ grid is considered as acceptable for a preliminary design study with lower computational costs of analyses. The analyses at this Reynolds number are within the laminar regime, and the use of turbulence models may have contributed to the error. Further analyses on this grid are performed at higher Reynolds numbers in the water-takeoff regime for amphibious aircraft, for which experimental data are not available; hence, this design point was used to validate the convergence study with available experimental data.

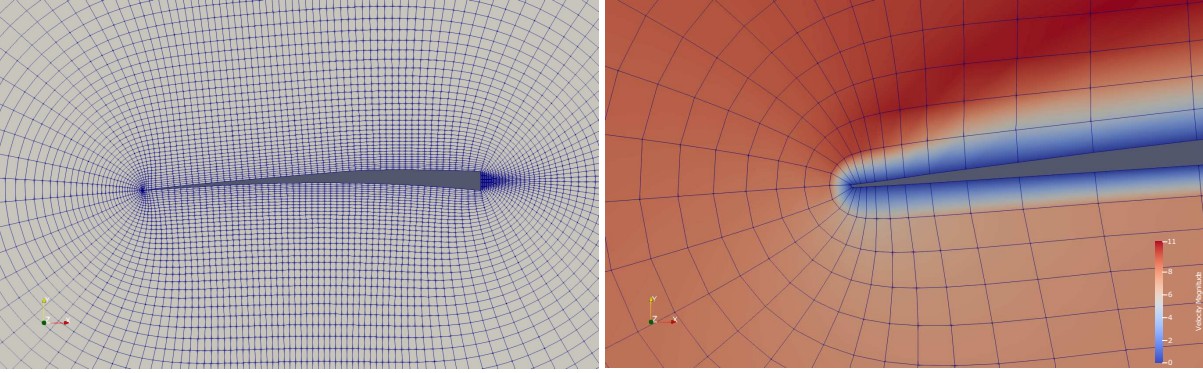

**Figure 8.** $L_1$ grid resolution and blunt leading edge with velocity contours (in m/s).

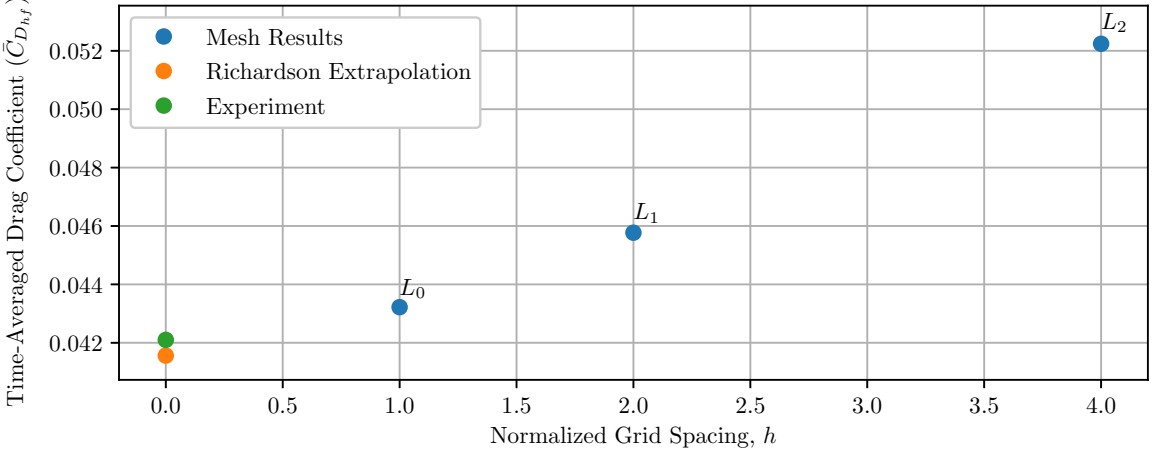

**Figure 9.** Grid convergence study via Richardson extrapolation.

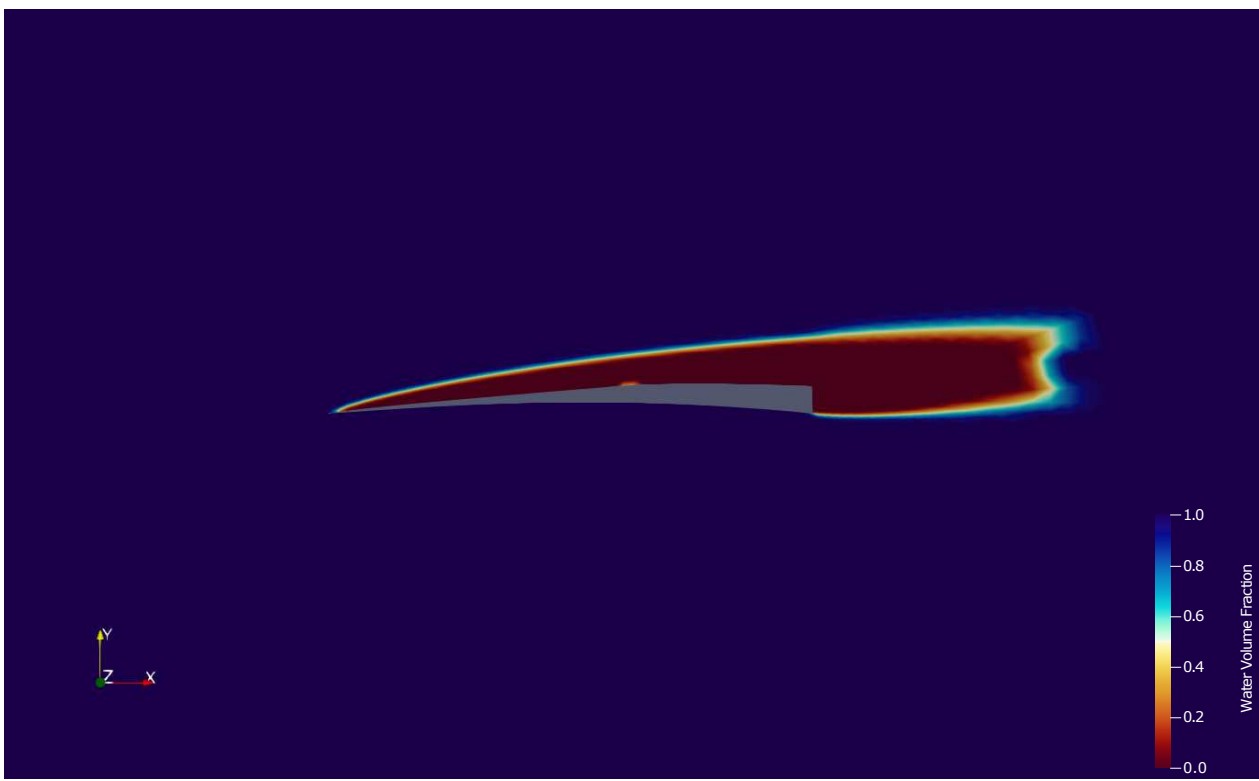

**Figure 10.** Supercavity over Waid-Lindberg profile.

**Table 4.** Grid convergence study and validation.

| Grid | Grid Size | $\overline{C}_{D_{hf}}$ | Error | $\overline{C}_{L_{hf}}$ | Error |
|------|-----------|-------------------------|-------|-------------------------|-------|
| Experiment | – | 0.0421 | 0% | 0.4199 | 0% |
| $L_2$ | 3950 | 0.0522 | 23.99% | 0.4667 | 11.14% |
| $L_1$ | 15,958 | 0.0458 | 8.79% | 0.4556 | 8.5% |
| $L_0$ | 63,832 | 0.0432 | 2.61% | 0.4480 | 6.69% |

4.2.2. Surrogate Models

The surrogate models for time-averaged $C_L$, $C_D$, and $C_M$ of the supercavitating hydrofoil are presented in Figure 11. Note that the profiles shown here significantly differ from ones generated by airfoils, as the cavitation effects are accounted for [1]. The surrogate models are generated using Kriging models via the SMT: Surrogate Modeling Toolbox [52] for the Waid-Lindberg hydrofoil profile using 25 sample points over $0 \le \alpha_{hf} \le 15$ and $2 \le u \le 45$. The 25 samples are selected from a data-set of uniformly distributed 100 points generated via the high-fidelity computational model mentioned in Section 3.4.2.2, i.e., the unsteady analyses with cavitation modeling. The timestep set for the CFD analyses was $2 \times 10^{-5}$ s, and the iterations indicated a maximum Courant-Friedrich-Lewys (CFL) number of 2 per timestep. The remaining 75 samples, denoted by $x_{s_i}$, are used for validation purposes, in which we compute the normalized root-mean square deviation (NRMSD), as shown in Equation (24). The errors are normalized by the ranges of the corresponding non-dimensional coefficients. The computed NRMSD values for $C_L$, $C_D$, and $C_M$ are tabulated in Table 5. These errors are deemed acceptable in approximating the coefficients.

$$\text{NRMSD} = \frac{1}{R} \sqrt{\frac{1}{N_s} \sum_{i=0}^{N_s} e_i^2}, \quad R = x_{\max} - x_{\min}, \ e_i = x_i - x_{s_i}, \quad x \in \{C_D, C_L, C_M\}. \quad (24)$$

The $(L/D)_{hf} = (C_L/C_D)_{hf}$ ratios, computed pointwise over the design space, are presented as a surface in Figure 12. The drastic reduction of $(L/D)_{hf}$ at higher speeds, which is caused by cavitation, is clearly visible in this profile.

**Table 5.** Error analysis of surrogate model for Waid-Lindberg supercavitating hydrofoil.

|  | $C_D$ | $C_L$ | $C_M$ |
|---|---|---|---|
| NRMSD, % | 2.8212 | 3.0168 | 3.8037 |

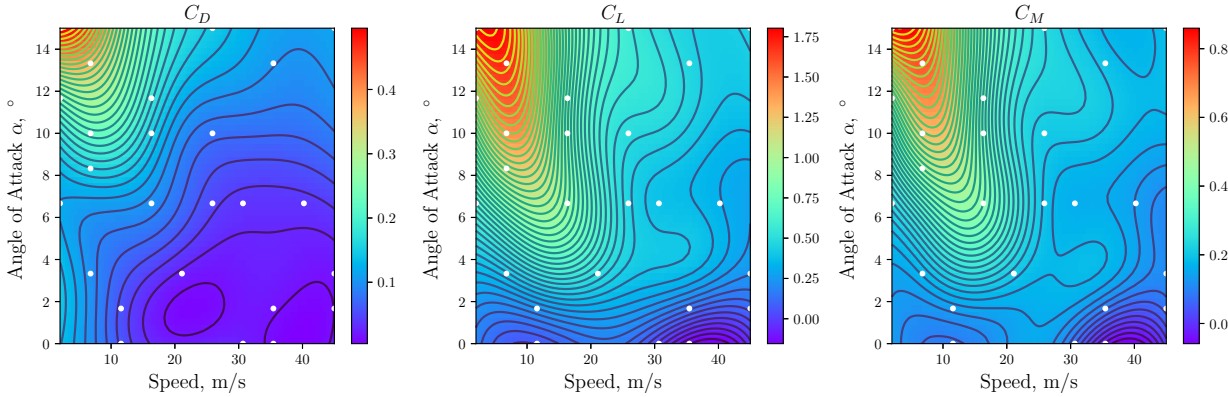

**Figure 11.** Surrogate contour plots for the coefficients of the Waid-Lindberg supercavitating hydrofoil. These profiles include the cavitation effects.

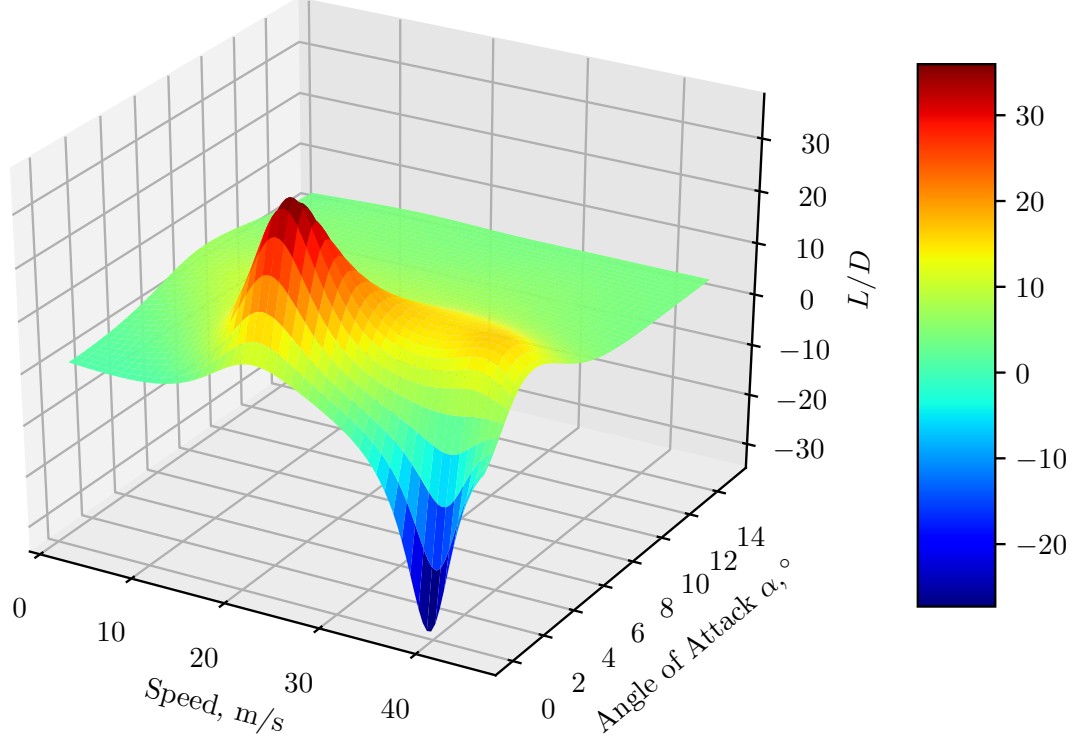

**Figure 12.** Response surface for the Waid-Lindberg supercavitating hydrofoil's $(L/D)_{hf}$ surrogate.

### 4.3. Optimization Studies

The optimizer used for the following studies is the Sequential Least-Squares Quadratic Programming algorithm provided via the SciPy package in Python [48]. This optimizer is suitable for optimization problems with bounds and constraints. The optimizers are set to converge at a tolerance of $10^{-12}$.

#### 4.3.1. Water-Takeoff Distance Minimization

A water-takeoff distance minimization study is performed to find the optimum span and incidence angle of the hydrofoil, following the procedure described in Section 3.1.3. As a retractable, non-foldable configuration is considered, the span is bounded from above by the hull beam width for the hydrofoil system to fit into the hull upon retraction.

The constrained design space with the baseline and optimum points are shown in Figure 13, in which dark dots with larger radii indicate design points in the feasible region at which the lift constraint is satisfied, and the colors corresponding to the colorbar depict the takeoff values for all points. This diagram also depicts the sensitivities of the objective function to the design variables. The number of iterations required to obtain convergence was 16, with 181 function evaluations and 16 gradient evaluations. The water-takeoff distance exhibits nonlinear behavior with respect to both variables, and the minimization problem appears to be multimodal and sensitive to initial guesses. Observing slices of fixed span lengths with varying incidence angles indicates that the takeoff distance sharply increases at specific incidence angle values. As the span length increases, the corresponding incidence angle decreases, and eventually extends to multiple incidence angles as can be seen in the region $2.2 \text{ m} \leq b_{hf} \leq 4.0 \text{ m}$, $0° \leq \alpha_{hf} \leq 4.0°$. This physically corresponds to the regions in which the hydrofoil generates large drag and insufficient lift to reduce the hull resistance effectively. At larger angles, such as the region between 6–14° for span lengths between $1 \text{ m} \leq b_{hf} \leq 6 \text{ m}$, the hydrofoil generates larger lift, hence reducing the takeoff distance values; however, this region violates the lift constraint and the optimizer chooses the lower takeoff distance values satisfying the constraint at the optimum shown in the diagram.

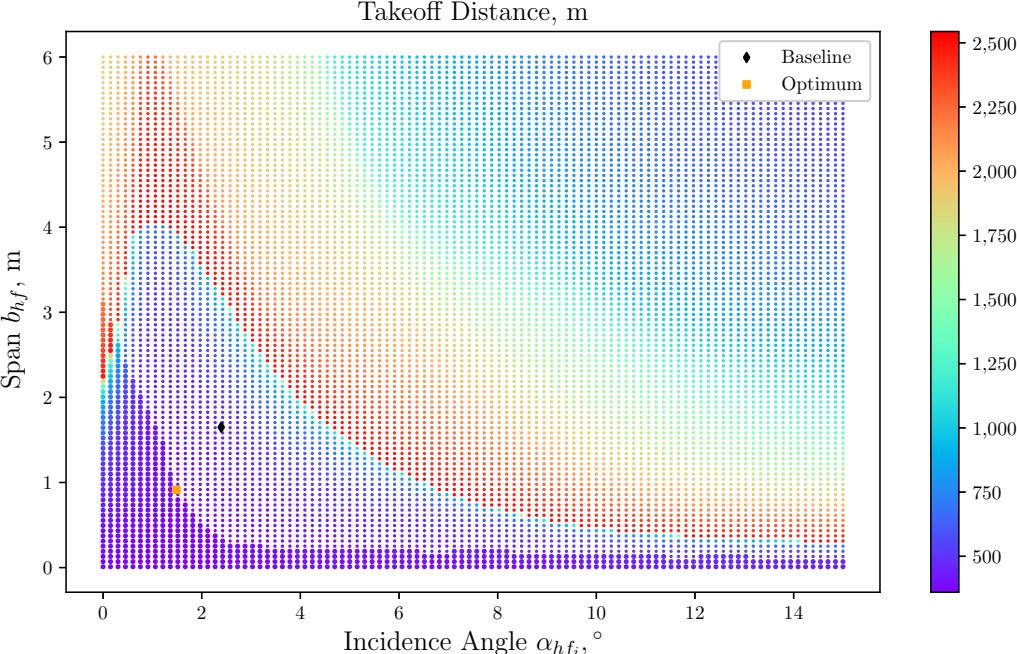

**Figure 13.** The design space for water-takeoff distance minimization to find the span and incidence angle of the hydrofoil. The darker region with larger radii depicts the area satisfying the lift constraint.

If an incidence angle of $\alpha_{hf_i} \geq 5°$ is selected as an initial guess with the initial span value, then the optimizer tends to miss the minimum at lower angles, as it tends towards higher angles. By choosing the right starting point, on the other hand, the global optimum within the bounds can be found more effectively. In particular, the initial guess is selected through an initialization process that aims to maximize the average $(L/D)_{hf}$. The maximum average $(L/D)_{hf}$ ratio over the takeoff speed range of the hydrofoil is evaluated as an initial guess to feed to the water-takeoff distance minimizer. This is done by evaluating the averages of the $(L/D)_{hf}$ diagram in Figure 12 over the speed range, providing the average $(L/D)_{hf}$ ratio as a function of the angle of attack of the hydrofoil. This procedure results in the graph shown in Figure 14. The maximum point obtained in this initialization procedure is then used to determine the incidence angle's initial value in the water-takeoff distance minimization procedure. A similar profile was obtained from a study on the YS-920 hydrofoil, designed for subcavitating conditions [1]. However, it suffered from lower $(L/D)_{hf}$ ratios and higher water-takeoff distances, indicating profiles designed for subcavitating regimes were not suitable for amphibious aircraft and further emphasized the need for using supercavitating hydrofoils in this particular application.

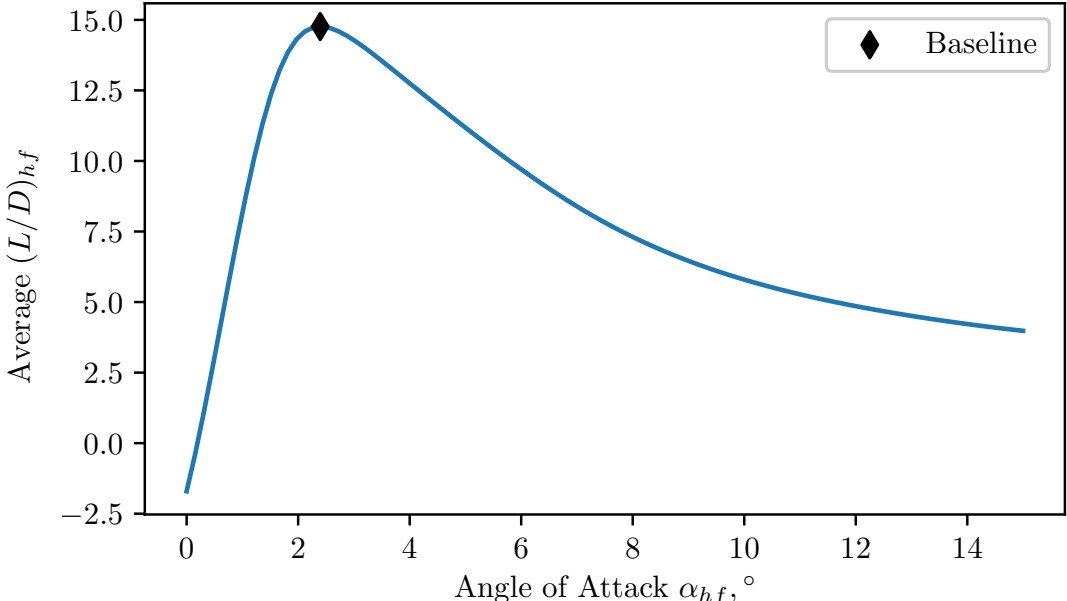

**Figure 14.** Average $(L/D)_{hf}$ Ratios for the Waid-Lindberg supercavitating hydrofoil.

### 4.3.2. Stabilizer Force Minimization

The optimum span and incidence angle are then provided to the stabilizer force minimization problem, to find the optimal hydrofoil position. The optimization procedure follows the description presented in Section 3.1.4. The initial values selected for the hydrofoil location are based on the criteria presented in [3] that the longitudinal position of the hydrofoil be located at approximately $0.5\bar{c}_w$ forward of the aircraft's CG. Its design space is depicted in Figure 15, which also depicts the sensitivities of the objective function to the design variables. It indicates that the stabilizer force is more sensitive to the horizontal location of the hydrofoil than its vertical location. It also indicates that the upper bound for $z_{hf}$, shown as the horizontal red line in the figure, will be the solution, and will calculate the appropriate $x_{hf}$ to minimize the stabilizer force, which is physically consistent with the considerations presented in Section 3.3.1. The number of iterations required to obtain convergence was 18, with 98 function evaluations and 18 gradient evaluations.

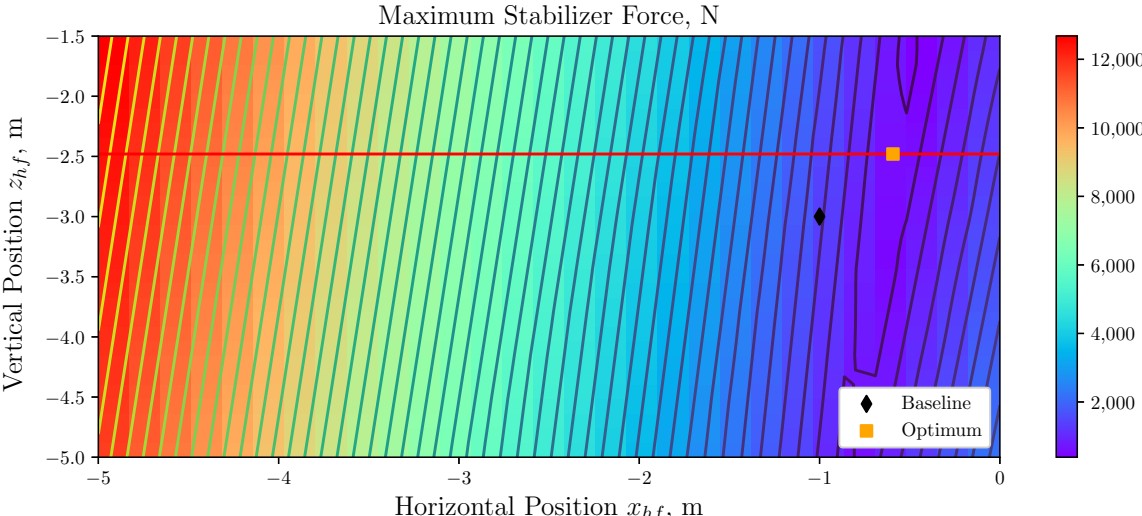

**Figure 15.** Hydrofoil position design space for the stabilizer force minimization. The red line indicates the upper bound of the hydrofoil's vertical position.

The results of both optimization procedures are summarized in Table 6. Detailed discussions on the performance comparison between the baseline and optimized designs are presented in the next section.

**Table 6.** Span length, incidence angle, and position optimization results.

| Optimization Problem | Parameter | Optimal Value |
|---|---|---|
| Water-Takeoff Distance | Distance, $x_W$ | 359 m |
| | Incidence angle, $\alpha_{hf_i}$ | 1.486° |
| | Span, $b_{hf}$ | 0.914 m |
| Maximum Stabilizer Force | Force, $|L_h|_{max}$ | 435 N |
| | Hydrofoil location, $(x_{hf}, z_{hf})$ | $(-0.5959 \text{ m}, -2.48 \text{ m})$ |

*4.4. Performance Evaluations, Comparisons, and Discussion*

In this section, we evaluate the performance of hydrofoils in the context of amphibious aircraft application. The water-takeoff analysis procedure described in Section 3.2, yields detailed information of force and load variations during takeoff, which provides insights into the effectiveness of adding hydrofoils.

First, we compare the water-takeoff performance of the aircraft with and without hydrofoils; note that the optimized hydrofoil configuration is considered in this comparison. The comparison of its effects on the hull resistance is shown in Figure 16. It indicates minor reductions in the wave resistance of the hull, and substantial reductions in the viscous resistance. These observations physically correspond to the hydrofoil reducing the submerged volume and wetted area during the water-takeoff by the hull being unported from water, as can be seen in the water-takeoff results shown in Figure 17, satisfying the purposes of the design. The viscous resistance of the configuration without hydrofoils indicates the hull is still submerged to some extent as it reaches the takeoff speed, whereas hydrofoils can help completely unport the hull before this speed. The optimization procedure of the incidence angle and span ensures that the lift constraint in Table 1 is satisfied; the hydrofoil's lift reduces as the wing's lift increases while the aircraft's speed is below the required takeoff speed, as observed in Figure 18.

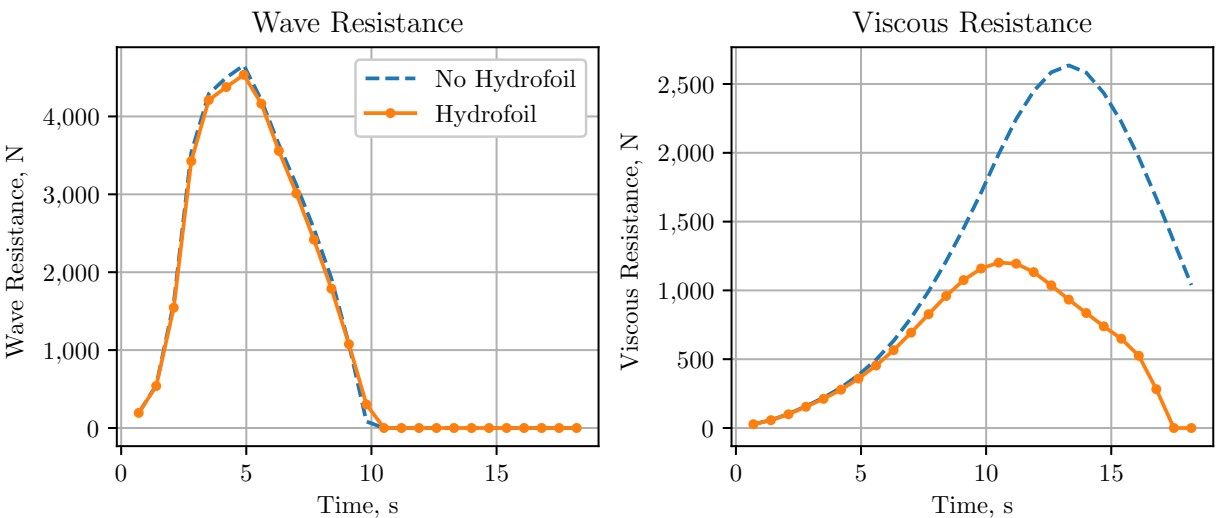

**Figure 16.** Hull resistance comparisons with and without the hydrofoil.

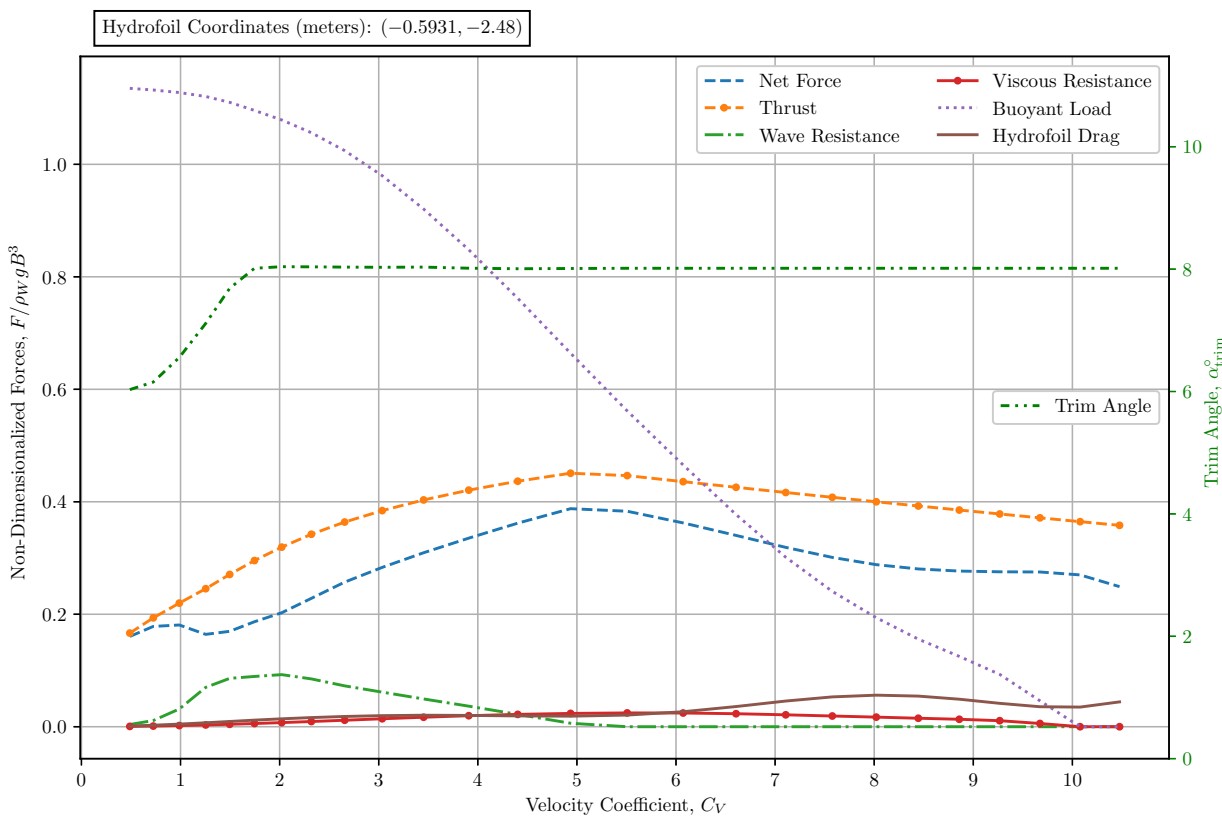

**Figure 17.** Variations of hydrodynamic and propulsive forces during water-takeoff process.

The results of the aircraft configuration with the baseline hydrofoil design (i.e., when using the initial values of the optimization design variables) are compared to those without and with the optimized hydrofoil design in Figure 19 to determine the performance differences and the relevant physics. A drawing depicting the differences between the baseline and optimized designs is presented in Figure 20. The comparisons between the three configurations in terms of water-takeoff distance and maximum stabilizer force are tabulated in Table 7. The notable reductions of both objective functions in the optimized design, compared to the baseline design, are evident in these results, which confirm the

effectiveness of the optimization procedure. More importantly, the results show that the optimized hydrofoils help reduce the water-takeoff distance by approximately 3.5% and the maximum stabilizer force by almost half as compared to those of the no-hydrofoil configuration, demonstrating the benefits of adding hydrofoils to amphibious aircraft. These results also emphasize the importance of performing optimizations, as adding hydrofoils based on physical reasoning alone does not guarantee performance improvement. The water-takeoff distance achieved by the aircraft configuration with the baseline hydrofoil, for instance, is longer than that of the no-hydrofoil configuration, which would defeat the purpose of adding hydrofoils.

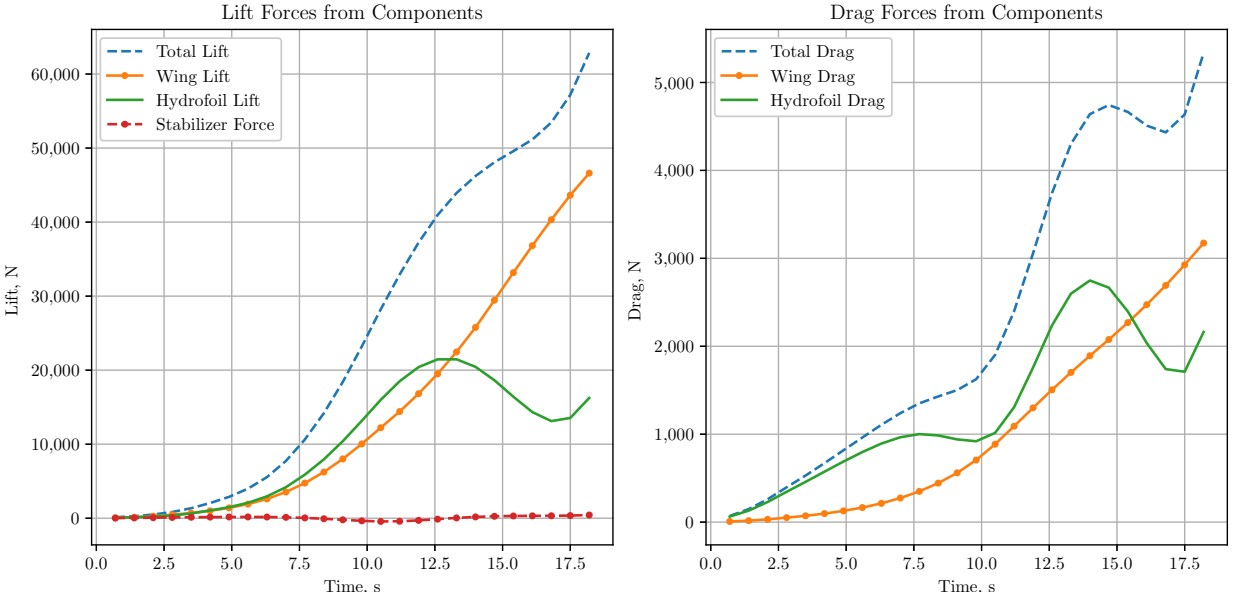

**Figure 18.** Lift and drag forces of components during water-takeoff process.

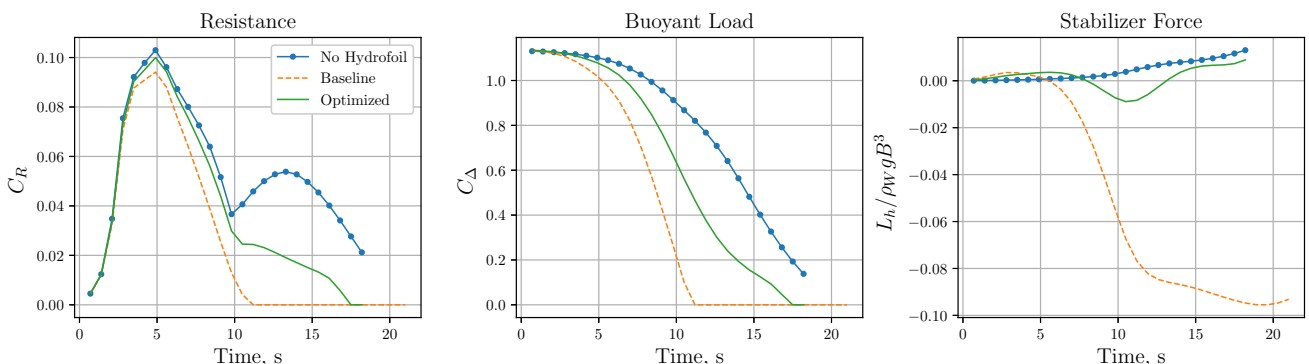

**Figure 19.** Force comparisons of the amphibious aircraft with and without the hydrofoil, considering both the baseline and optimized hydrofoil configurations.

The baseline design achieves the aim of reducing the buoyant load, shown in Figure 19, before the no-hydrofoil configuration, but it generates excessive moments that may not be corrected by the elevator, based on the required stabilizer force also seen in Figure 19. The relatively lower absolute values of the stabilizer force required for the optimized design as compared to the baseline design indicate the optimization procedure for the hydrofoil location is appropriate. Physically, the optimized design exhibits growing sinusoidal oscillations of the required stabilizer force, indicating a porpoising tendency for the aircraft during the water-takeoff if the trim corrections are not performed.

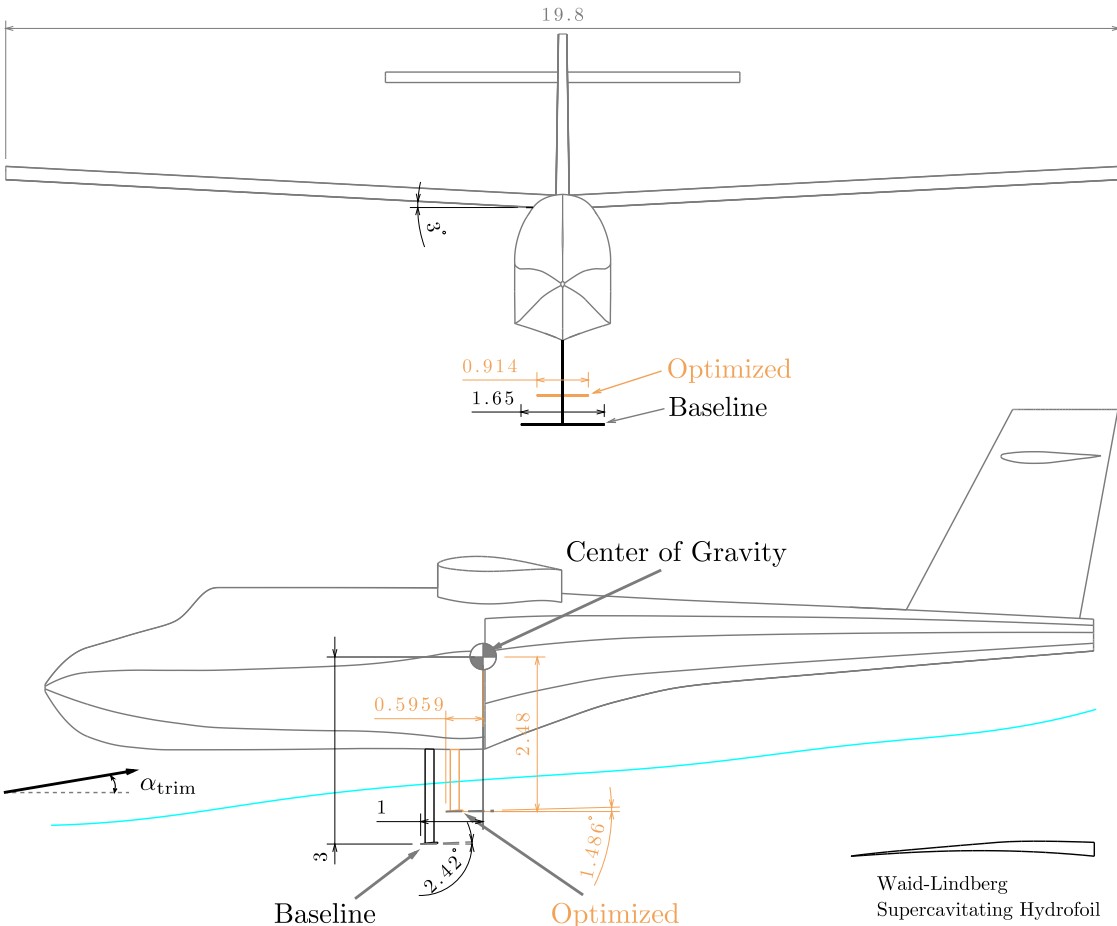

**Figure 20.** Aircraft drawing with baseline and optimized hydrofoil configurations (in meters).

**Table 7.** Water-takeoff performance results and comparisons between the different configurations.

|  | No Hydrofoil | Baseline | Optimized |
|---|:---:|:---:|:---:|
| Water-Takeoff Distance, m | 372 | 461 | 359 |
| Maximum Stabilizer Force, N | 604 | 4743 | 435 |

## 5. Conclusions

In the present study, the key considerations for the preliminary design of a hydrofoil for amphibious aircraft were investigated. A sizing procedure was formulated with recommendations for profile selection and different configurations. In particular, we observed that a supercavitating hydrofoil profile was required for amphibious aircraft application. An optimization framework was formulated to minimize the water-takeoff distance of the aircraft by finding the optimal span length and incidence angle. Its detrimental effects on the longitudinal stability of the aircraft with respect to the required horizontal stabilizer force to maintain the trim angle were also minimized, by finding the optimal position of the hydrofoil with respect to the aircraft's center of gravity. The effectiveness of the design was analyzed by a water-takeoff physics simulator using surrogate models of the hydrofoil. These models were generated via CFD sample data to determine the performance of a selected hydrofoil over a range of speeds and angles of attack within the water-takeoff regime below the takeoff speed.

To evaluate and assess the performance and effectiveness of adding hydrofoils to amphibious aircraft, the developed preliminary design framework was applied in a 10-seater aircraft as a case study. Results from the water-takeoff simulator showed the variations of forces and loads during the water-takeoff run, providing insights into the physics and

behavior of the aircraft during the process. The aircraft's porpoising tendency, for instance, was evident by observing the stabilizer variations, which highlighted the importance of trim corrections. These results were unique to amphibious aircraft, i.e., the hydrofoil performance was different from that of marine applications, and the takeoff performance was different from that of conventional ground-based aircraft.

The results of the water-takeoff procedure indicated that the hydrofoil performed the required purposes of reducing hull resistance and buoyant load. These reductions demonstrated the hydrofoil's effectiveness in improving the water-takeoff performance and efficiency of amphibious aircraft, as evident in the reduction of the water-takeoff distance observed in the case study. Our results showed, however, that adding hydrofoils alone, without being properly optimized, could not guarantee a performance improvement. This observation further emphasized the need to develop a comprehensive design framework for amphibious aircraft and highlighted the benefits of performing optimization in design studies. The optimization results indicated that the water-takeoff distance minimization with respect to the span and incidence angle as design variables was non-differentiable and multimodal with the required constraints, depending on the hydrofoil's profile. The problem of minimizing the stabilizer force was relatively less complex, but still an important part of the design procedure.

To the best of our knowledge, ours is the first study that systematically evaluates hydrofoil performance in the context of amphibious aircraft operation, particularly during the water-takeoff process. The modular design of the framework gives flexibility to replace the current models, be it aerodynamic, hydrodynamic, or propulsive, with higher-fidelity models in future development. The computation with the surrogate models allows for rapid analysis with variations of parameters, taking less than two minutes to complete for a case, which offers a desirable computational efficiency. This computational framework will reduce reliance on expensive experiments which have limited the technological advancement and progress in amphibious aircraft development. More importantly, the developed framework will enable performing detailed amphibious aircraft design optimization and tradeoff studies with more computational rigor. We will be able to perform multidisciplinary design optimization that simultaneously considers the aerodynamic and hydrodynamic performance (e.g., to optimize hull and hydrofoil shapes) to obtain a truly optimal design, thereby advancing the amphibious aircraft development.

**Author Contributions:** Conceptualization, A.S. and R.P.L.; methodology, A.S.; software, A.S.; validation, A.S.; formal analysis, A.S.; investigation, A.S.; resources, R.P.L.; data curation, A.S.; writing—original draft preparation, A.S.; writing—review and editing, R.P.L.; visualization, A.S.; supervision, R.P.L.; project administration, R.P.L.; funding acquisition, R.P.L. All authors have read and agreed to the published version of the manuscript.

**Funding:** This research was funded by the HKUST Initiation Grant (Grant No. R9354).

**Institutional Review Board Statement:** Not applicable.

**Informed Consent Statement:** Not applicable.

**Data Availability Statement:** The data presented in this study are available on request from the corresponding author. The data are not publicly available due to ongoing research with further refinements for future studies.

**Acknowledgments:** The authors would like to thank the HKUST Post-Graduate International Student Fellowship Scheme for partially funding the first author.

**Conflicts of Interest:** The authors declare no conflict of interest.

## Abbreviations

The following symbols used in the manuscript are provided here for reference:

| | |
|---|---|
| $(x, z)$ | 2D coordinates with respect to center of gravity as origin |
| $t$ | Time |
| $\tau$ | Non-dimensional time |
| $\overline{(\,)}$ | Time-averaged quantity |
| $\Delta(\,)$ | Change in quantity |
| $\dot{(\,)}$ | Time derivative of quantity |
| $g$ | Acceleration due to gravity on Earth |
| $h$ | Reference height for hydrostatic pressure head |
| $u$ | Speed in horizontal direction |
| $p_\infty$, $p_{\text{atm}}$, $p_V$ | Freestream, atmospheric and vapor pressure |
| $\rho$ | Density of fluid |
| $\eta$ | Dynamic pressure ratio |
| $\mu$ | Dynamic viscosity of fluid |
| $Re$, $We$, $Fr$, $Ca$ | Reynolds, Weber, Froude, and cavitation number |
| $(\,)_w$, $(\,)_h$, $(\,)_{hf}$ | Quantity corresponding to wing, horizontal tail, and hydrofoil |
| $L$, $D$, $M$ | Lift, drag and moment |
| $C_L$, $C_D$, $C_M$ | Lift, drag, and moment coefficients |
| $C_p$, $C_F$ | Pressure and skin-friction resistance coefficients |
| $C_\Delta$, $C_R$, $C_V$ | Load, resistance and speed coefficients |
| $\alpha$, $\alpha_i$, $\alpha_{\text{trim}}$ | Angle of attack, incidence, and trim angle |
| $L/D$ | Lift-to-drag ratio |
| $V_c$, $Z_c$ | Longitudinal stability coefficients of component $c$ |
| $l_h$, $l_v$ | Horizontal and vertical moment arms |
| $b$, $c$ | Span and chord lengths |
| $S$ | Reference area |
| $L$ | Reference length |
| $AR$ | Aspect ratio |
| $W$ | Aircraft weight |
| $B$ | Hull beam width |
| $\delta$ | Dihedral angle |
| $\bar{c}$ | Mean aerodynamic chord length |
| $(\bar{x}, \bar{z})$ | Coordinates non-dimensionalized with respect to mean aerodynamic chord |
| $T$ | Thrust |
| $\phi_T$ | Thrust angle |

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
