# Peer review of "Amphibious Aircraft Developments: Computational Studies of Hydrofoil Design for Improvements in Water-Takeoffs†"

_aerospace, doi:10.3390/aerospace8010010_

Round 1

Reviewer 1 Report

The work concerns the problem of the optimal selection of hydrofoils parameters for a seaplane to minimize the take-off distance on the water and minimize the horizontal stabilizer force.

Seaplanes have again attracted a lot of attention as a potential alternative means of transport to other air and non-air transport modes, particularly attractive in island areas or areas with developed coastline or a large number of lakes. Seaplanes do not require airport infrastructure, so their development is not limited by the existence of airports. They can use both seaports and river ports.

Designing seaplanes is usually a much more difficult process than designing land planes. In the case of the fuselage, the requirements of hydrodynamics and aerodynamics, as well as flight and water stability must be considered. Determining the take-off characteristics also involves more complex analysis, taking into account aerodynamic and hydrodynamic phenomena that change depending on the speed. The use of experimental data on phenomena such as cavitation and ventilation, obtained in the study of ships, is possible only to a small extent because these data are obtained at a constant speed of the ship, and the seaplane moves at a variable speed.

The authors analysed various design solutions of seaplanes, selecting a seaplane equipped with hydrofoils for analysis. Classical seaplane design methods, described in the literature, are also discussed.

For the selected configuration, the authors stated that the way to shorten the water take-off distance of a seaplane equipped with hydrofoil is to go to planning as quickly as possible.

The work is not a description of classic seaplane conceptual design. The authors of the study focused only on the optimization of hydrofoils geometry, without analysing the influence of other design parameters. The authors used surrogate models obtained based on low-fidelity and high-fidelity analysis, which allowed to significantly shorten the computation time without a significant decrease in their accuracy. The optimization was carried out for the criterion of the shortest possible take-off distance on the water.

The research problem has been formulated clearly and transparently. The methodology was formulated correctly and completely. The obtained results have been described briefly, but sufficiently.

One of the problem is that there is no list of symbols used. Most of the symbols are described in the text, but it requires constant searching for their definition in various parts of the text. Also, a drawing defining the hydrofoil's geometry and its position in concerning to other seaplane parts would greatly help in better understanding the problem.

As the solution is quite sensitive to hydrofoil parameters, it might be worth showing an additional analysis of the solution's sensitivity to changing the optimized parameters.

Reviewer 2 Report

The paper is characterized by reliability, transparency and comprehensive discussion of the solved optimisation problem in the field of aeronautical engineering. I do not raise any major reservations neither to the substantive nor formal (language, graphics, a division into chapters, etc.) side of the paper.

However, I have a few suggestions for possible additions that, in my opinion, could contribute to a more legible and more convincing presentation of the results presented by the Authors:

  1. It would be good to add a three-dimensional (but simplified) drawing showing a seaplane (e.g. DHC-6 TWIN OTTER) in three configurations:
    - No Hydrofoil
    - Baseline
          - Optimized
    The drawing should clearly show the essential differences between the three configurations, important from point of view of the optimisation goals and results.
  2. In any optimization process, the extent to which the "Optimized" configuration is "better" than the "Baseline" depends heavily on the selection of the "Baseline". It would be good for the Authors to convince us that they chose a typical configuration as "Baseline" and not a configuration with particularly "bad properties".

3. Although it is not explicitly stated in the text, it is probably the case that the comparison of the seaplane properties in the following configurations:
      - No Hydrofoil
      - Baseline
      - Optimized
has been carried out on the basis of a simplified calculation methodology, the same as used in the optimisation process (i.e. stationary, 2D or quasi-3D calculations, use of surrogate models, etc.). The presented optimisation results would be significantly more probable if high-fidelity, three-dimensional, unsteady CFD simulations were performed for the above three seaplane configurations and the results of these simulations were attached to the article.

Reviewer 3 Report

The following reports show essential experience with early hydrofoil supported seaplanes, an approch of such a theory; but even ventilation and cavitation testing of seaplane hydrofoil profiles and systems could be of interest:

General A. Guidoni: "Seaplanes - Fifteen Years of Naval Aviation", Journal of the Royal Aeronautical Society, Vol. 32 (No. 205), Jan. 1928

F. Weinig: "Zur Theorie des Unterwassertragflügels und der Gleitfläche", Luftfahrtforschung (Band 14), Lfg. 6 (in German) - ("On the theory of hydrofoil and planing surface")

W. Sottorf: "Der Wassertragflügel in Anwendung auf das Seeflugzeug",Jahrbuch 1937 der deutschen Luftfahrtforschung, Bd. I (in German) - ("The hydrofoil in use with seaplane") cf. to Abb.12 - Abb. 14.

W. Sottorf: "Experimentelle Untersuchungen zur Frage des Wassertragflügels", FB 1319, Deutsche Luftfahrtforschung, Institut f. Seeflugwesen, DVL, Berlin-Adlershof/Hamburg, 13.12.1940 (in German) - ("Experimental investigations on the question of hydrofoil") - gives the results of systematic and comprehensive investigations on circular segment profiles with regard to seaplane hydrofoil profiles and configurations. Based on the circular segment profile, the influence of thickness, curvature, S-shape with flat and hollow pressure side and nose shape on lift, resistance, flow course, splash formation and cavitation ingress is investigated. In a comparative study close to the water surface, a sharp-edged thin circular segment profile with increasing curvature in the leading edge area of the suction side is determined as the most favorable hydrofoil profile in every aspect. The influence of aspect ratio, diving depth and keel angle is recorded. The outlook of the hydrofoil and the upper speed limit for use with seaplanes are discussed.

It would be helpful for the reader to insert schematic drawings of the different hydrofoil configurations like surface-piercing hydrofoils, foldable-protruding hydrofoils, and strut-based hydrofoils for better understanding.

By the way the old Dornier company had tested hydrofoils on the Do S flying boat at the beginning of the 1930ies.
